# TXNIP loss expands Myc-dependent transcriptional programs by increasing Myc genomic binding

**Tian-Yeh Lim, Blake R. Wilde[¤a], Mallory L. Thomas, Kristin E. Murphy[¤b], Jeffery M. Vahrenkamp, Megan E. Conway, Katherine E. Varley, Jason Gertz, Donald E. Ayer***

Department of Oncological Sciences, Huntsman Cancer Institute, Salt Lake City, Utah, United States of America

¤a Current address: Department of Biological Chemistry, David Geffen School of Medicine, University of California, Los Angeles, Los Angeles, California, United States of America
¤b Current address: Department of Biomedical Genetics, Wilmot Cancer Institute, University of Rochester Medical Center, Rochester, New York, United States of America
* don.ayer@hci.utah.edu

**Data Availability Statement:** Raw and processed RNA-seq, ChIP-seq and ATAC-seq data has been deposited in the Gene Expression Omnibus under

## Abstract

*The c-Myc* protooncogene places a demand on glucose uptake to drive glucose-dependent biosynthetic pathways. To meet this demand, c-Myc protein (Myc henceforth) drives the expression of glucose transporters, glycolytic enzymes, and represses the expression of thioredoxin interacting protein (TXNIP), which is a potent negative regulator of glucose uptake. A Myc$_{high}$/TXNIP$_{low}$ gene signature is clinically significant as it correlates with poor clinical prognosis in triple-negative breast cancer (TNBC) but not in other subtypes of breast cancer, suggesting a functional relationship between Myc and TXNIP. To better understand how TXNIP contributes to the aggressive behavior of TNBC, we generated TXNIP null MDA-MB-231 (231:TKO) cells for our study. We show that TXNIP loss drives a transcriptional program that resembles those driven by Myc and increases global Myc genome occupancy. TXNIP loss allows Myc to invade the promoters and enhancers of target genes that are potentially relevant to cell transformation. Together, these findings suggest that TXNIP is a broad repressor of Myc genomic binding. The increase in Myc genomic binding in the 231:TKO cells expands the Myc-dependent transcriptome we identified in parental MDA-MB-231 cells. This expansion of Myc-dependent transcription following TXNIP loss occurs without an apparent increase in Myc's intrinsic capacity to activate transcription and without increasing Myc levels. Together, our findings suggest that TXNIP loss mimics Myc overexpression, connecting Myc genomic binding and transcriptional programs to the nutrient and progrowth signals that control TXNIP expression.

## Introduction

Thioredoxin interacting protein (TXNIP) is an α-arrestin protein with several anti-proliferative functions, predominant among these are activities as a negative regulator of glucose

accession numbers GSE208412, GSE208415 and GSE222694.

**Funding:** This work was funded by National Institutes of Health grants (www.nih.gov), R01CA222650 (DEA) and R01 HG008974 (JG), and by 132596-RSG-18-197-01-DMC from the American Cancer Society (www.cancer.org) (KEV). Funds from Huntsman Cancer Institute's National Cancer Institute Cancer Center Support Grant (P30CA042014) and the Huntsman Cancer Foundation (https://healthcare.utah.edu/huntsmancancerinstitute/foundation/) also provided support. The funders had no role in study design, data collection and analysis, decision to publish, or preparation of the manuscript.

**Competing interests:** The authors have declared that no competing interests exist.

**Abbreviations:** ChIP-qPCR, chromatin immunoprecipitation quantitative PCR; DMEM, Dulbecco's Modified Eagle Medium; FBS, fetal bovine serum; GSEA, Gene Set Enrichment Analysis; HDR, homology-directed repair; HOMER, Hypergeometric Optimization of Motif EnRichment; LC–MS, liquid chromatography–mass spectrophotometry; MACS2, Model-Based Analysis of ChIP-seq-2; METABRIC, Molecular Taxonomy of Breast Cancer International Consortium; PCR, polymerase chain reaction; RT-qPCR, reverse transcription-quantitative PCR; siMyc, siRNA Myc-targeting; siNT, siRNA non-targeting; TGF-β, transforming growth factor beta; TNBC, triple-negative breast cancer; TSS, transcriptional start site; TXNIP, thioredoxin interacting protein.

uptake and as a suppressor of multiple progrowth signaling pathways [1–5]. Consistent with a role in restricting cell growth, TXNIP expression is suppressed by multiple cancer-associated progrowth signaling pathways [6–9], and its expression is typically low in tumors compared to adjacent normal tissue. Further, TXNIP expression decreases with increasing tumor grade, and low TXNIP expression is correlated with poor clinical outcomes in several cancers [10–15]. Whether low TXNIP expression supports cell growth by increasing glucose utilization, activating progrowth pathways or whether other mechanisms also contribute is currently unknown.

Our previous work demonstrated that low TXNIP expression correlates with poor clinical outcomes in TNBC, but not in other breast cancer subtypes [12]. Further, this correlation is more pronounced in patients with elevated expression of the Myc transcription factor, suggesting functional interaction(s) between TXNIP and Myc. Their crosstalk occurs at least at 2 levels. First, Myc drives expression of many glucose-dependent biosynthetic pathways [16–22], suggesting that low TXNIP expression (high glucose uptake) in combination with high Myc expression (high glucose use) helps cells match glucose availability with glucose utilization. Second, Myc represses TXNIP expression by displacing the MondoA transcriptional activator from a shared E-box element located just upstream of the TXNIP transcriptional start site (TSS) [12]. Together, these findings suggest a feedforward mechanism where Myc's repression of TXNIP increases glucose uptake to support Myc-driven and glucose-fueled biosynthetic pathways.

Myc is implicated in more than 50% of human malignancy with elevated levels of transcriptionally active Myc being critical for its oncogenic function [23]. Elevated Myc levels arise from many mechanisms including, but not limited to, transcriptional and translational mechanisms and protein stability [24,25]. Myc drives transcription as a heterodimer with Max, with Myc levels being limiting for the formation of Myc:Max complexes [26]. The model that has emerged over the last several years is that at physiological levels, Myc:Max complexes bind to high affinity E-box sequences in the promoters and enhancers of genes involved in housekeeping pathways that support cell growth, such as ribosomal biogenesis [23]. At oncogenic levels, Myc:Max complexes invade lower affinity sites in the promoters and enhancers of genes that are associated with processes critical to cellular transformation, such as signaling pathways. Thus, increasing Myc levels expands the Myc-dependent transcriptome rather than simply elevating the expression of Myc-dependent transcripts [27–30].

In this report, we provide evidence that TXNIP loss drives a global increase in Myc genomic binding and drives gene expression programs enriched for known Myc targets. Surprisingly, this expansion of the Myc transcriptome was not accompanied by an increase in Myc protein expression, suggesting that TXNIP loss increases in Myc's activity as a transcription factor. Our data support a model where TXNIP loss increases Myc's capacity to bind its genomic targets, rather than increasing its intrinsic capacity to activate or repress transcription.

## Results

### TXNIP loss mimics Myc overexpression

To better understand how TXNIP contributes to the aggressive behavior of TNBC, we used CRISPR/Cas9 editing to delete TXNIP from MDA-MB-231 cells (231:TKO) (Figs 1A and S1A). We chose MDA-MB-231 cells for our analysis because they have intermediate levels of Myc and TXNIP mRNA expression (S1B Fig). TXNIP deletion does not result in changes in Myc protein expression (Figs 1B and S1C), nor does it alter the rate of cell proliferation (S1D Fig). To understand the broad transcriptional changes that arise following TXNIP loss, we performed RNA-seq on 231:TKO and unedited MDA-MB-231 cells (parental 231). Using cutoffs

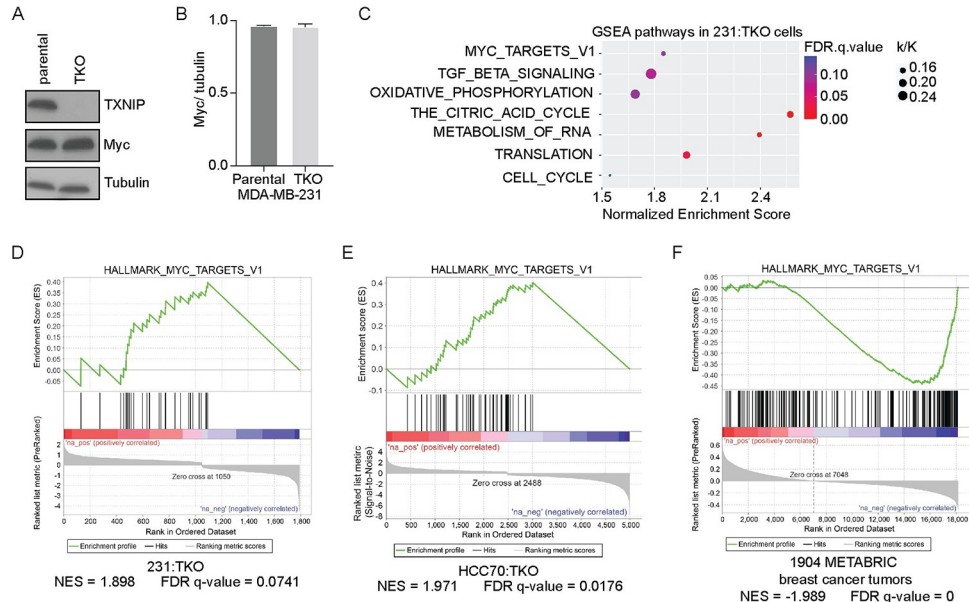

**Fig 1. TXNIP loss mimics Myc overexpression.** (**A**) Western blotting was used to determine the levels of the Myc, TXNIP, and tubulin in MDA-MB-231 (parental 231) and 231:TKO cells. (**B**) Quantification of Myc protein levels (normalized to those of tubulin) from (**A**) in parental 231 and TKO cells using ImageJ. The image of the western blot used for quantification is shown in S1C Fig. (**C**) RNA sequencing was performed on 2 biological replicates of each parental 231 and 231:TKO cells to identify TXNIP-regulated genes. Preranked GSEA was performed by comparing up- and down-regulated genes in 231:TKO cells with the Hallmark and Reactome datasets in the MSigDB. The enriched GSEA pathways of TXNIP-regulated genes were plotted using ggplot2 package from R studio. The k/K value is a ratio of number of genes in our data set (k) divided by the number of genes in the indicated dataset (K). (**D**) The preranked GSEA plot of enrichment of the regulated genes in 231:TKO cells with the Hallmark_Myc_targets_v1 dataset. Because the FDR q-value accounts for multiple hypothesis testing and the small size of the datasets being queried, FDR q-values of up to 0.25 are suitable for hypothesis generation [31,32]. (**E**) The preranked GSEA of regulated genes in HCC70: TKO (HCC:TKO) cells with the Hallmark_Myc_targets_v1 dataset. (**F**) A rank ordered gene list was developed by correlating TXNIP expression with the expression levels of all transcripts expressed in the 1,904 breast cancer tumors available in the METABRIC dataset [34]. This preranked gene list was used in a GSEA of the Hallmark datasets from the MSigDB. An enrichment plot for Hallmark_Myc_targets_v1 is shown. The underlying data for Fig 1B and 1C can be found in S1 Data. GSEA, Gene Set Enrichment Analysis; METABRIC, Molecular Taxonomy of Breast Cancer International Consortium; MSigDB, Molecular Signatures Database; TKO, TXNIP-knockout; TXNIP, thioredoxin interacting protein.

of reads >5 counts, and an adjusted *p*-value (pAdj) ≤0.05, we identified 1,050 and 742 genes whose expression was up- and down-regulated, respectively, in response to TXNIP deletion. INHBB, KISS1, FOXA2, and ZNF704 were among the most highly down-regulated genes, whereas AKR1C3, MT-ATP8, and G0S2 were the most highly up-regulated genes (S1E Fig).

To identify TXNIP-regulated pathways, we compared our 231:TKO dataset with annotated gene sets using preranked Gene Set Enrichment Analysis (GSEA) [31,32]. Given that Myc levels and proliferation rate were not altered by TXNIP loss, it was surprising that the TXNIP-null dataset was positively enriched for known Myc targets (Fig 1C and 1D). Other pathways enriched in the 231:TKO cells included pathways involved in transforming growth factor beta (TGF-β) signaling, oxidative phosphorylation, the citric acid cycle, metabolism of RNA, translation, cell cycle, the G2M checkpoint, and fatty acid metabolism (Figs 1C and S1F). The identification of Myc targets and pathways known to be regulated by Myc in the 231:TKO cells raises the possibility that TXNIP loss mimics Myc overexpression, yet these changes in gene expression apparently occur with no discernable change in Myc protein expression. Further, as TXNIP loss does not affect proliferation of MDA-MB-231 cells, we suggest that the

increased Myc transcriptional programs are not simply an epiphenomenon downstream of increased cell division.

To determine the generality of TXNIP loss on Myc-like gene signatures, we deleted it from 2 additional cell lines. First, we deleted TXNIP from HCC70 TNBC cells, which also have intermediate levels of Myc and TXNIP mRNA (S1B Fig), and performed RNA-seq. Similar to TXNIP loss in MDA-MB-231 cells, TXNIP loss did not increase Myc levels or proliferation rates in HCC70 cells (S2A–S2C Fig), yet Myc transcriptional targets were also up-regulated following TXNIP deletion (Fig 1E). Second, we deleted TXNIP from immortalized human myoblast MB135 cells (MB135:TKO) [33]. TXNIP loss did not increase Myc levels or cell proliferation rates in MB135 cells (S2D–S2F Fig). We differentiated the parental MB135 and MB135:TKO cells into myotubes for 5 days and used RNA-seq to determine their transcriptional profiles. Similar to the MDA-MB-231 and HCC70 cells, differentiated MB135:TKO cells were enriched for known Myc targets (S2G Fig). Together, these data show that TXNIP loss results in the up-regulation of known Myc targets in multiple TNBC cell lines and in an immortal myoblast cell line, suggesting that TXNIP may be a general regulator of Myc transcriptional programs and that this activity is not restricted to transformed cells.

To determine whether the inverse relationship between TXNIP and Myc transcriptional programs is restricted to cell lines, we downloaded the gene expression data from 1,904 breast tumors annotated in the Molecular Taxonomy of Breast Cancer International Consortium (METABRIC) dataset [34]. We then calculated Pearson coefficients for every measured gene correlated to TXNIP expression. We next generated a ranked gene list using the Pearson correlation coefficients and used this list to perform GSEA. This ranked dataset was negatively enriched with a geneset containing known Myc targets (Fig 1F). This finding suggests that TXNIP-correlated gene expression programs in breast cancers are inversely correlated with known Myc-dependent transcriptional programs. TXNIP-correlated gene expression programs were also negatively enriched with several other progrowth datasets, including mTOR signaling, E2F targets, and additional Myc targets (S3A Fig), and positively enriched in several datasets including inflammatory responses, apoptosis, and adipogenesis (S3B Fig). Together, our data suggest that TXNIP is capable of regulating Myc transcriptional programs, not only in cell lines, but in bona fide breast tumors as well.

## Gene expression changes in 231:TKO cells are Myc dependent

Because TXNIP loss in MDA-MB-231 cells leads increased the expression of known Myc targets, we next determined whether the changes in gene expression in the 231:TKO cells were Myc dependent. To do so, we reduced Myc levels using a short interfering RNA approach and performed RNA-seq on RNA isolated 48 hours after transfection of the siRNAs. We achieved robust Myc knockdown (Fig 2A). Further, proliferation rates were not affected during the time course of the Myc knockdown (S4A Fig), suggesting that changes in gene expression may be attributed to reduced Myc expression rather than to a change in proliferation rate. The RNA-seq analysis revealed 5,669 transcripts that were up- and down-regulated in 231:TKO +siMyc cells compared to TKO:231+siNon-Targeting (siNT) control cells using a pAdj < 0.05. As expected, known Myc transcriptional programs were strongly down-regulated 231:TKO +siMyc cells (S4B Fig). The 231:TKO+siMyc dataset was also negatively enriched for other pathways that were also up-regulated in the 231:TKO cells, e.g., E2F targets and pathways involved in metabolism of RNA, protein translation, and cell cycle (Fig 2B). In general, the enriched pathways shown in Fig 2B are the most highly enriched pathways across the various experiments presented in this manuscript. For simplicity and clarity, we present the same group enriched datasets in each figure that contains this type of data. Of the 1,792 transcripts

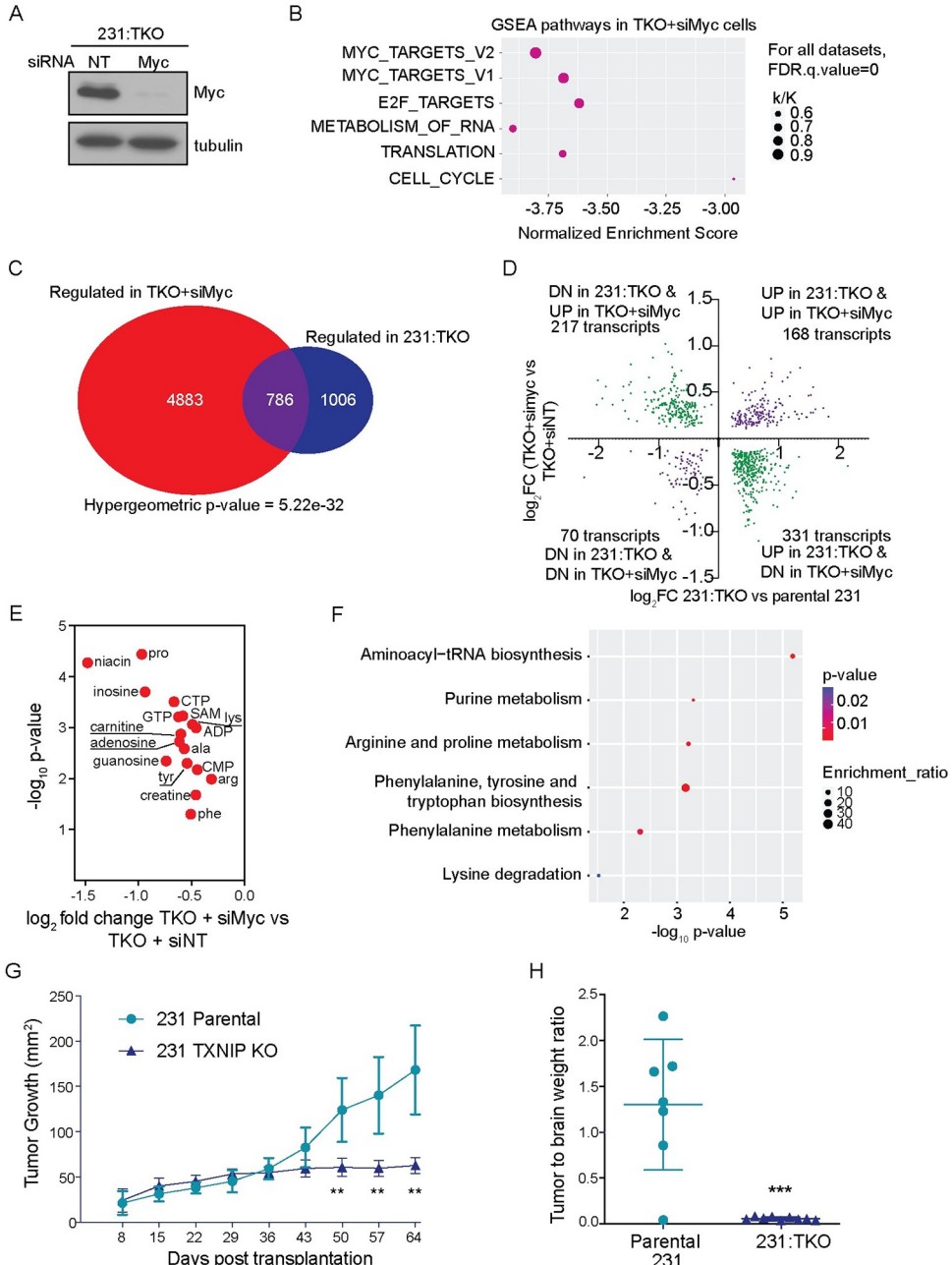

**Fig 2. Gene expression changes in TKO are Myc dependent.** (**A**) Western blotting was used to determine c-Myc and tubulin levels in 231:TKO cells following siRNA-mediated Myc knockdown. siRNA non-targeting control: siNT or siRNA targeting Myc: siMyc. (**B**) RNA sequencing was performed on 3 biological replicates 231:TKO with siNT or 231:TKO cells with siMyc to identify Myc-dependent genes in 231:TKO cells. Preranked GSEA was performed to identify pathways for Myc-dependent genes by comparing a ranked list of up- and down-regulated genes in 231:TKO +siMyc cells with the Hallmark and Reactome datasets in the MSigDB. The enriched GSEA pathways of Myc-dependent genes in 231:TKO cells were plotted using ggplot2 package from R studio. The k/K value is a ratio of number of genes in our data set (k) divided by the number of genes in the indicated datasets (K). (**C**) All regulated genes with adjusted *p*-value < 0.05 in 231:TKO were compared with all regulated genes with adjusted *p*-value < 0.05 in 231:TKO+siMyc dataset to identify genes regulated in both datasets. The Venn diagram was drawn using a VennDiagram package in R studio. (**D**) The 786 Myc-dependent transcripts were subdivided into 4 categories based on the direction of their regulation in the 231:TKO and 231:TKO+siMyc datasets. (**E**) Seventeen metabolites that were up-regulated in 231:TKO cells also showed down-regulation with Myc knockdown in TKO cells. (**F**) Pathway analysis was performed using MetaboAnalyst to identify the metabolic pathways that were reciprocally regulated by Myc and TXNIP. The enriched metabolic pathways were plotted using ggplot2 package from R studio. (**G**) Parental 231 and 231:

TKO cells were transplanted into the cleared mammary fat pads of NOD scid immunocompromised mice. Tumor growth was determined using caliper measurements at the indicated times. Shown is a representative of 2 independent experiments. Values are reported as means with standard deviation. **$p < 0.01$ (**F**) After 64 days of in vivo growth, tumor weight relative to brain weight was determined using $t$ tests. ****$p < 0.0001$. The underlying data for Fig 2B and 2D–2F can be found in S1 Data. GSEA, Gene Set Enrichment Analysis; MSigDB, Molecular Signatures Database; TKO, TXNIP-knockout; TXNIP, thioredoxin interacting protein.

that were regulated in the 231:TKO cells, about 43% (786 genes) were dependent on Myc (Fig 2C). Of these 786 genes, 548 transcripts (approximately 70%) were reciprocally regulated in 231:TKO and 231:TKO+siMyc cells, suggesting that TXNIP and Myc have primarily opposing functions in regulating gene expression (Fig 2D). Pathway analysis suggests that this group of reciprocally regulated genes largely account for the Myc signatures identified in the 231:TKO +siMyc cells (S4C Fig). Ribosomal protein genes are well-established Myc targets [35], and they exemplify the reciprocal relationship between TXNIP and Myc. For example, TXNIP loss results in up-regulation of 24 ribosomal protein genes encoding proteins from both the large and small ribosomal subunit (approximately 30% of the 79 ribosomal protein genes) and all 24 of these genes were down-regulated following Myc depletion (S4D Fig). Collectively, these data show that TXNIP loss generates gene expression programs that resemble Myc-driven gene expression programs. Further, these gene expression programs are highly Myc dependent.

To begin to understand the functional outcome of the interplay between TXNIP and Myc, we next used mass spectrometry approaches to measure steady state metabolites in 231:TKO cells before and after Myc knockdown. TXNIP loss resulted in the up-regulation of multiple metabolites and the levels of the majority of these were reduced following Myc knockdown using an siRNA approach (Fig 2E). These metabolites were enriched in progrowth metabolic pathways such as tRNA, purine, and amino acid biosynthesis (Fig 2F). Thus, the reciprocal relationship between TXNIP and Myc at the level of gene expression is also evident at the level of progrowth metabolic pathways.

Finally, to test the effect of TXNIP loss on orthotopic tumor growth in vivo, we transplanted parental 231 and 231:TKO cells into the cleared mammary fat pads of immunocompromised mice and used caliper measurements to monitor their growth over time. In this setting, we found, for approximately the first 5 weeks of the experiment, parental 231 and parental 231: TKO cells grew at the same rate. After this time, the parental 231 xenografts continued to grow; however, the growth of the 231:TKO cells slowed dramatically (Fig 2G). At the termination of the experiment, the tumors from the parental 231 cells were significantly larger than those from the 231:TKO cells (Fig 2H). Together, these data suggest that while TXNIP loss does not alter the in vitro growth of several cell lines (S1D, S2C and S2F Figs), its expression is strictly required for growth of orthotopic xenografts.

## TXNIP loss regulates global Myc genomic binding

TXNIP loss resulted in increased Myc-dependent gene expression programs, yet did not result in broad up-regulation of gene expression across the genome. These findings suggest that TXNIP loss might increase Myc transcriptional programs by a direct mechanism, rather than by an indirect mechanism that would derepress transcription genome-wide. Because TXNIP loss does not change Myc levels (Figs 1A, 1B, and S1C), we performed Myc ChIP-seq on parental 231 and 231:TKO cells to assess whether TXNIP regulates global Myc binding. We identified about 5,600 Myc-occupied binding sites in parental 231 cells and roughly 28,000 sites in 231:TKO cells (q-value cutoff = 0.01) (Fig 3A), suggesting that TXNIP is a broad repressor of Myc genomic binding. After filtering out binding sites with a percentage difference greater than 60%, we used the kmeans function in deepTools to identify 3 clusters of Myc

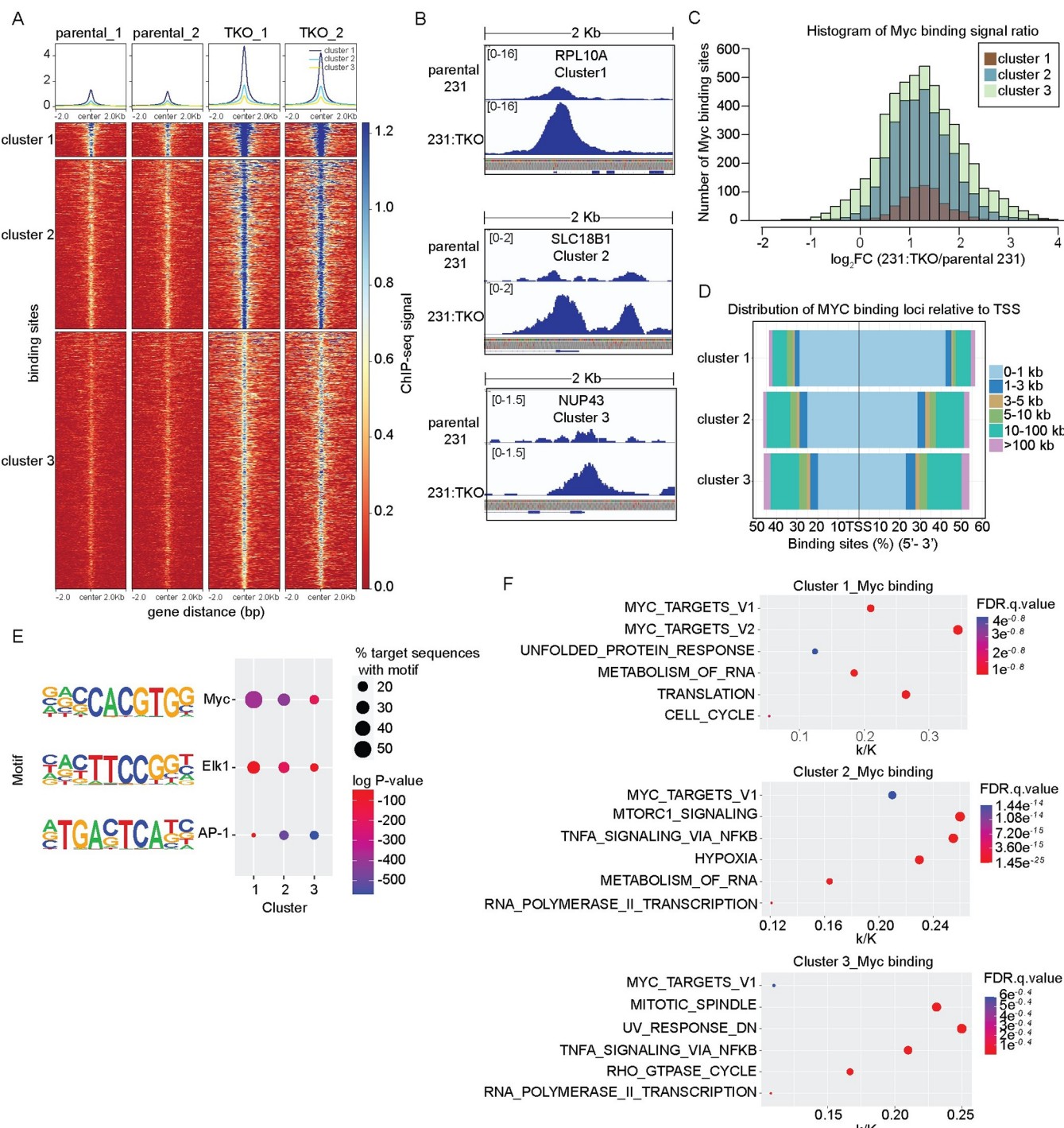

**Fig 3. TXNIP regulates global Myc genomic binding.** (**A**) A heatmap of Myc ChIP-sequencing data of 2 biological replicates of each parental 231 and 231: TKO cells was divided into 3 clusters using deepTools with the clustering argument of kmeans. (**B**) Myc binding, as visualized using IGV_2.5.2, to selected genes from each cluster in parental 231 and 231:TKO cells. (**C**) Histogram showing the Myc occupancy signal ratio of Myc binding sites in each cluster. Myc binding ratio was calculated by dividing the counts in 231:TKO with the counts in parental 231 cells. (**D**) The distance of Myc binding sites from TSS in each cluster was annotated using the ChIPseeker program. (**E**) Enriched sequence motifs in the proximity of Myc-occupied sites in the 3 clusters were determined using HOMER. (**F**) Myc-binding events were associated with potentially regulated genes using ChIPseeker. This set of Myc-associated genes were then evaluated for their enrichment in the Hallmark or Reactome datasets in MSigDB using GSEA. k/K value is a ratio of number of genes in our data set (k) divided by the number of genes in the indicated dataset (K). The underlying data for Fig 3C and 3D can be found in S1 Data. GSEA, Gene Set Enrichment Analysis; HOMER, Hypergeometric Optimization of Motif EnRichment; MSigDB, Molecular Signatures Database; TKO, TXNIP-knockout; TSS, transcriptional start site; TXNIP, thioredoxin interacting protein.

binding sites. In parental cells, Myc genomic binding was highest at sites in cluster 1 (669 sites), intermediate at sites in cluster 2 (3,333 sites), and lowest at sites in cluster 3 (5,063 sites). Myc binding in each cluster increased dramatically in 231:TKO cells. Genome browser views of representative sites in each cluster, RPL10A (cluster1), SLC18B1 (cluster 2), and NUP43 (cluster 3) showed a clear increase in Myc binding in 231:TKO cells as expected (Fig 3B). We also determined the fold increase or decrease in Myc binding (Myc signal ratio) in the 231: TKO cells for sites in each of the 3 clusters (Fig 3C). This analysis revealed that (1) approximately 96% of Myc binding sites showed elevated Myc binding in 231:TKO cells; (2) that the majority of Myc binding sites showed a slightly more than a 2-fold increase in Myc occupancy in the 231:TKO cells; and (3) that the increase in Myc signal ratio was similar at the binding sites in each cluster. Together, these data demonstrate that TXNIP loss leads to a global increase in Myc binding and support the hypothesis that TXNIP is a repressor of Myc genomic binding.

We used ChIPseeker [36] to associate Myc genomic binding events with specific genes and identified 644, 2,680, and 3,974 genes in clusters 1, 2, and 3, respectively. In cluster 1, 71.0% of the Myc binding sites were located within +/− 1 kilobase (kb) of the TSS of the associated genes. By contrast, the percentage of genes with TSS-proximal Myc binding sites progressively decreased in clusters 2 and 3, with a concomitant increase in Myc binding events at sites located between 10 and 100 kb from the TSS (Fig 3D).

We used Hypergeometric Optimization of Motif EnRichment (HOMER) [37] analysis to identify sequence elements enriched close to the Myc binding site in each of the 3 clusters. This analysis revealed that canonical CACGTG Myc-binding motifs were associated with roughly 50% of the Myc binding events in cluster 1, with the percentage of the canonical sites decreasing to 26% and 17% in clusters 2 and 3, respectively. We also discovered Elk1/ETS and AP-1 motifs enriched close to the Myc-binding peak in all 3 clusters, with Elk1/Ets motifs decreasing from cluster 1 to 3 and AP1 motifs increasing (Fig 3E). Together, these data suggest that cluster 1 contains the highest affinity canonical Myc binding sites that are located primarily in the promoters of the associated genes. Further, the data suggest that clusters 2 and 3 contain lower affinity Myc binding sites that diverge from canonical Myc binding element and are located more distal to the TSS, perhaps in regulatory enhancers.

Because previous publications indicate that Myc can regulate different subgroups of targets based on the affinity of the Myc binding site [27,28], we used the MSigDB [31,32] to identify pathways enriched for Myc-binding sites in each cluster. This analysis revealed established Myc targets in each cluster with the highest percent enrichment in the Myc_Targets_V1 geneset in clusters 1 and 2 with a k/K value of 0.21. The enrichment of this dataset decreased in cluster 3 with k/K value of 0.11. The Myc_Targets_V2 geneset was only enriched in cluster 1 with a k/K value of 0.34. Genesets associated with translation and the metabolism of RNA, including genes encoding several ribosomal proteins, were enriched in clusters 1 and 2. mTORC1 and hypoxia-associated genesets were strongly enriched in cluster 2, whereas TNFA_signaling via NFKB, mitotic_spindle, and Rho_GTPase_cycle genesets were highly enriched in cluster 3 (Fig 3F). Together, these data demonstrate that different sets of target genes are enriched in each cluster and, consistent with previous reports [27,28], suggest that Myc occupancy of different sets of target genes appears to be dictated by Myc's differential binding to regulatory elements of these targets.

## G0S2 is reciprocally regulated by TXNIP and Myc

We next investigated the Myc- and TXNIP-dependent regulation of G0S2 as representative of their reciprocal function in gene regulation. We chose to examine G0S2 for 3 reasons: (1) it

was among the most highly up-regulated genes in the 231:TKO cells (S1E and S5A Figs); (2) a previous publication showed that G0S2 was up-regulated in the livers of TXNIP knockout mice [38]; and (3) G0S2 in an inhibitor of triglyceride breakdown, which may contribute to the high levels of triglycerides observed in TXNIP knockout mice [39,40]. We first confirmed that G0S2 mRNA and protein were up-regulated in 231:TKO cells (Fig 4A and 4B). Our Myc

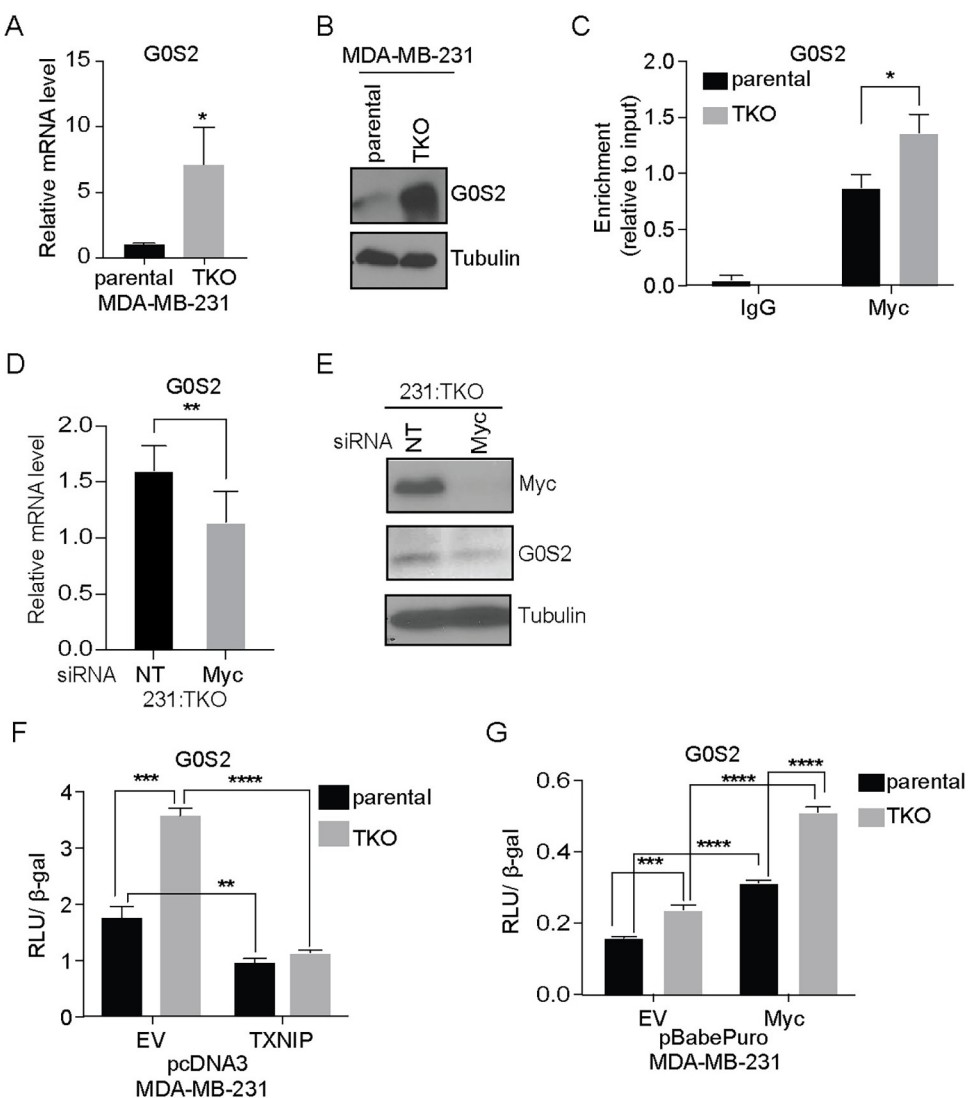

**Fig 4. G0S2 is reciprocally regulated by TXNIP and Myc.** (**A**) Human G0S2 (G0S2) mRNA levels in parental 231 and 231:TKO cells were measured using RT-qPCR. (**B**) Levels of G0S2 and tubulin in parental 231 and 231:TKO cells were determined using western blotting. (**C**) We performed Myc ChIP-qPCR to measure Myc occupancy upstream of the G0S2 TSS using 3 biological replicates of parental 231 and 231:TKO cells. Statistical significance was determined using a $t$ test. $^*p < 0.05$. (**D** and **E**) G0S2 mRNA (**D**) and protein (**E**) levels in 231:TKO cells were measured using RT-qPCR and western blotting following siRNA-mediated Myc knockdown for 48 hours. (**F**) Luciferase activities from a G0S2 luciferase reporter in lysates from parental 231 and 231:TKO cells with ectopic human TXNIP expression from pcDNA3 vector or pcDNA3 EV were measured. Luciferase activity was normalized to the β-gal activity. (**G**) Luciferase activities from a G0S2 luciferase reporter in lysates from parental 231 and 231TKO cells with ectopic human Myc expression from pBabePuro vector or pBabePuro EV were measured. For F and G, values are reported as mean and standard deviation. $^{**}p < 0.01$; $^{***}p < 0.001$, $^{****}p < 0.0001$. The underlying data for Fig 4A, 4C, 4D, 4F, and 4G can be found in S1 Data. EV, empty vector; RT-qPCR, reverse transcription-quantitative PCR; TKO, TXNIP-knockout; TSS, transcriptional start site; TXNIP, thioredoxin interacting protein; β-gal, beta-galactosidase.

ChIP-seq experiment showed that Myc binding increased at the G0S2 promoter just upstream of the G0S2 TSS in the 231:TKO cells (S5B Fig). The Myc binding site in the G0S2 promoter was in cluster 2, suggesting that it is a medium affinity binding site. We confirmed that Myc binding increased at the G0S2 promoter following TXNIP loss using a ChIP-PCR approach (Fig 4C). Myc knockdown in 231:TKO cells reduced the levels of G0S2 mRNA and protein (Fig 4D and 4E), demonstrating that Myc is necessary for the elevation of G0S2 expression observed in 231:TKO cells. Together, these data suggest that TXNIP loss drives increased Myc binding to the G0S2 promoter resulting in its increased expression.

To investigate the role of the G0S2 promoter in mediating the effects of TXNIP loss further, we generated a luciferase reporter construct that contains 1,493 base pairs upstream of the G0S2 translational start site that encompass the Myc-occupied region identified in our ChIP experiments (S5B Fig). This reporter was more active in 231:TKO cells mirroring the effects of TXNIP of G0S2 expression when expressed from its endogenous promoter (Fig 4F). Further, reporter activity was diminished by TXNIP overexpression. Myc overexpression increased the activity of the G0S2 reporter (Fig 4G), demonstrating that Myc is sufficient to drive G0S2 expression from this minimal reporter construct. Finally, we recapitulated our finding that TXNIP is a repressor of G0S2 expression using a luciferase reporter comprised of the rat G0S2 promoter (S5C Fig) [41]. Importantly, mutation of a double E-box motif in the rat promoter, which is analogous in position and sequence to the Myc binding site in the human G0S2 promoter (S5B Fig), diminished reporter activity in both parental 231 and 231:TKO cells (S5D and S5E Fig). Collectively, these data support a model where TXNIP loss leads to increased Myc binding to the double E-Box element in the G0S2 promoter resulting in Myc-dependent activation of G0S2 expression.

## TXNIP loss increases Myc binding to drive Myc-dependent gene signatures

To examine the relationship between the Myc-dependent gene expression programs and increased Myc binding in 231:TKO cells, we compared the Myc-dependent transcripts identified in 231:TKO+siMyc cells with genes that showed increased Myc occupancy in 231:TKO cells. We found that 2,903 (51.2%) Myc-dependent genes identified in 231:TKO+siMyc cells showed increased Myc binding in the 231:TKO cells (Fig 5A). These 2,903 genes were enriched in similar pathways as those enriched in 231:TKO+siMyc cells (S6A Fig). By contrast, we identified 4,996 Myc binding sites that were not associated with changes in Myc-dependent gene expression in 231:TKO cells, suggesting that many Myc-binding events did not lead to measurable changes in gene expression. We examined 2 pathways with Myc-regulated genes in more detail. We found that 72.4% (21/29) and 87.5% (21/24) of the Myc-dependent transcripts enriched in the Myc_Targets_v1 gene set and genes encoding ribosomal proteins, respectively, showed elevated Myc binding close to the TSS (Fig 5B and 5C). Interestingly, most of the Myc_Targets_v1 had binding sites within 1 kb of the TSS, whereas the proximity of the Myc binding site to the TSS of the ribosomal protein genes was more mixed with some genes having Myc binding sites within 1 kb of the TSS, with others having sites more distant. In contrast to these genesets, transcripts in the oxidative phosphorylation geneset that were regulated by TXNIP loss, showed less Myc dependence and fewer Myc-binding events (S6B Fig). These results suggest that TXNIP loss increases Myc-dependent gene expression by increasing Myc genome occupancy.

To better understand what constitutes a functional Myc binding event, we evaluated additional parameters. We found no correlation between the number of Myc sites and the magnitude of Myc-transcriptional regulation in either down- or up-regulated genes in 231:TKO +siMyc cells (S6C and S6D Fig). Further, the distance of the Myc-binding site relative to the

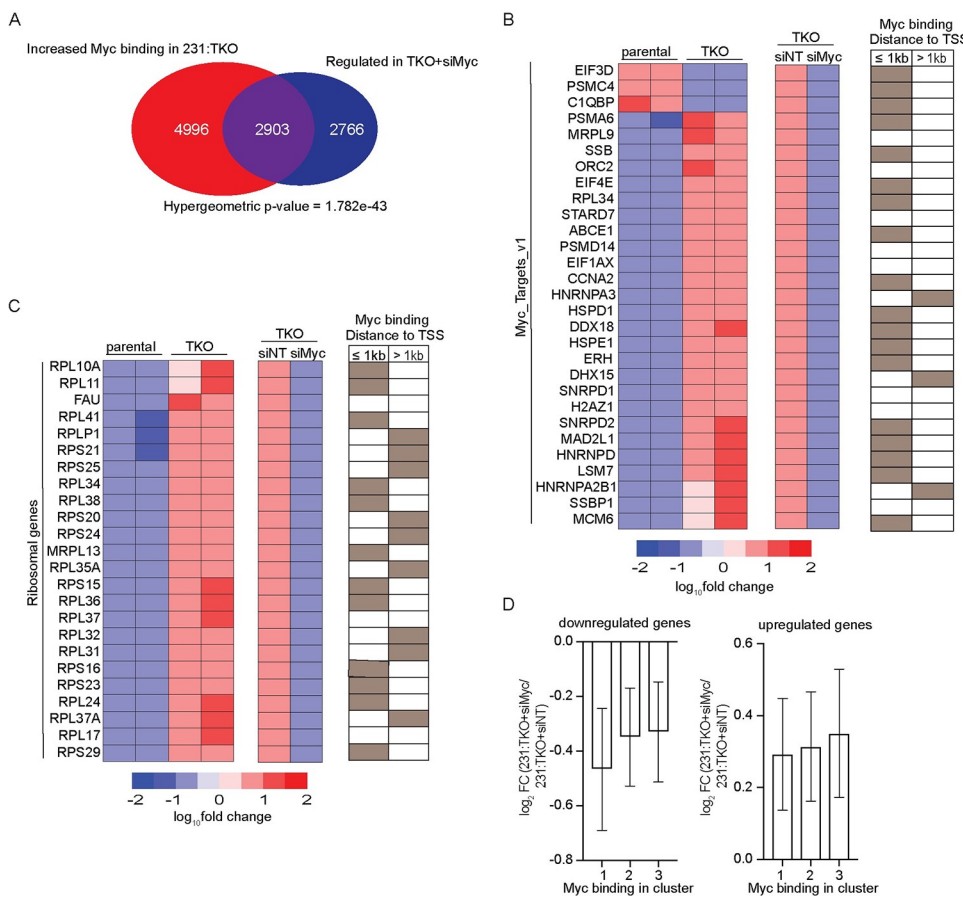

**Fig 5. TXNIP loss drives Myc-dependent gene expression programs.** (**A**) Myc-dependent genes in 231:TKO cells were compared with genes with increased Myc occupancy in 231:TKO cells. (**B** and **C**) Heatmaps of regulated genes in 231:TKO cells and in 231:TKO+siMyc cells that are enriched in Hallmark Myc_Targets_v1 (**B**) and in Reactome ribosomal protein genes (**C**) were plotted. The distances of Myc binding sites from TSS are determined using GREAT [43]. Myc binding sites less than 1 kb or more than 1 kb from TSS are indicated by brown boxes. Open boxes indicate no Myc binding. (**D**) The $\log_2$ fold change ($\log_2$ FC) of down-regulated and up-regulated genes in 231:TKO+siMyc cells compared to 231:TKO+siNT cells were plotted versus Myc binding events in each of the 3 Myc-binding clusters. The underlying data for Fig 5D can be found in S1 Data. GREAT, Genomic Regions Enrichment of Annotations Tool; TKO, TXNIP-knockout; TSS, transcriptional start site; TXNIP, thioredoxin interacting protein.

TSS of a regulated gene did not correlate with the magnitude of Myc regulation. Approximately 60% of Myc-activated (i.e., down-regulated in 231:TKO+siMyc) genes had a Myc binding site within 1 kb of the TSS. By contrast, only 30% of the Myc-repressed genes (i.e., up-regulated in 231:TKO+siMyc) had a Myc binding site within 1 kb of the TSS (S6E and S6F Fig). Although more Myc-activated genes had Myc binding sites closer to the TSS than Myc-repressed genes, there was no correlation between the degree of Myc regulation and the distance to the Myc binding sites for either Myc-activated or Myc-repressed genes loci. Finally, there was no relationship between the magnitude of Myc dependence and whether the Myc binding site(s) associated with the regulated gene were present in cluster 1, 2, or, 3 (Figs 3A and 5D). Thus, we observed that about 50% of the Myc-regulated genes had an associated Myc-binding event; however, there was no apparent relationship between the number of Myc binding sites, the affinity of those sites or the distance of the Myc binding site from the TSS, and the magnitude of Myc dependence.

To understand whether the global Myc biding events in 231:TKO cells are conserved across other cell lineages, we compared our genomic Myc binding profiles with those from the U2OS osteosarcoma cell line [27] and the Ramos Burkitt's lymphoma cell line [42]. We identified roughly 30,000 Myc binding sites in each cell line with about 1/3 of the Myc-binding sites (11,145 sites) identified in all 3 cell types (S7A Fig). About 85% of the common Myc binding sites were located within 1 kb of the TSS of the associated gene (S7B Fig) and were highly enriched for canonical CACGTG Myc-binding E-boxes (S7C Fig). The 11,145 shared Myc sites were associated with 7,871 genes, of which 39% (3,070 genes) showed Myc-dependent transcription in 231:TKO cells (S7D Fig). The 3,070 genes were strongly enriched of known Myc-dependent transcriptional programs (S7E Fig). Together, these data suggest that we have identified a conserved set of Myc-binding sites and Myc-dependent transcriptional targets in 231:TKO cells.

## TXNIP loss expands the Myc transcriptome

TXNIP loss led to a dramatic increase in global Myc binding and the induction of Myc-dependent gene expression signatures (Figs 1–3), suggesting that its loss fundamentally altered the Myc-dependent transcriptome. To investigate whether TXNIP loss might also increase Myc-dependent transcriptional activity, we used an siRNA approach to knock Myc down in parental 231 cells and determined differentially expressed transcripts using RNA sequencing (Fig 6A). The knockdown of Myc in parental 231 cells was as robust as it was in the 231:TKO cells, reducing Myc levels below the level of detection by western blotting. Inspection of the RNA-seq data also revealed that Myc mRNA levels were reduced about 2-fold in each cell line by Myc knockdown (S8A Fig). As with Myc knockdown in the 231:TKO cells, reduction of Myc in the parental 231 cell did not affect proliferation during the time course of the experiment (S8B Fig). Given the similarities in the magnitude of Myc knockdown the parental 231 and 231:TKO cells and the lack of effect on proliferation following Myc knockdown, we believe that gene expression differences between parental 231and 231:TKO cells are attributable to bona fide changes in gene expression rather than secondary effects.

We identified 1,196 genes that were Myc dependent (pAdj < 0.05) in parental 231 cells. By comparison, there were about 5 times as many Myc-dependent transcripts in 231:TKO cells (5,669 transcripts) (Figs 2C and 6B). Most of the Myc-dependent transcripts identified in the parental 231 cells (1,045 transcripts/approximately 87%) were also Myc dependent in the 231:TKO cells (Fig 6B). As expected, the Myc-dependent genes in both the parental 231 and 231:TKO cells were negatively enriched in Myc targets, E2F targets, and pathways involved in RNA metabolism, translation, and the cell cycle (Fig 6C); however, in 231:TKO cells, these gene sets had lower normalized enrichment scores and increased overlap ratio (k/K) compared to parental 231 cells (Fig 6C). These data suggest that while TXNIP loss increased the number of Myc-dependent transcripts compared to parental 231 cells, it does not generally alter the biological pathways regulated by Myc. One notable exception is that 231:TKO cells appear to be enriched for genes encoding transcripts involved in oxidative phosphorylation (NES = −1.329, FDR q-value = 0.1734). Thus, TXNIP loss appears to expand the Myc-dependent transcriptome.

We next wondered whether the 1,045 Myc-dependent transcripts common to parental 231 and 231:TKO cells were regulated similarly by Myc. Supporting this notion, the magnitude of Myc dependence of the shared 1,045 genes as a group was not significantly different in the 2 cell populations (Fig 6D). Further, the expression of groups of transcripts encoding proteins of similar function, i.e., solute carrier proteins, ribosomal proteins, and cell cycle regulators, also showed a similar degree of Myc dependence in the 2 cell populations (S8C Fig). Finally,

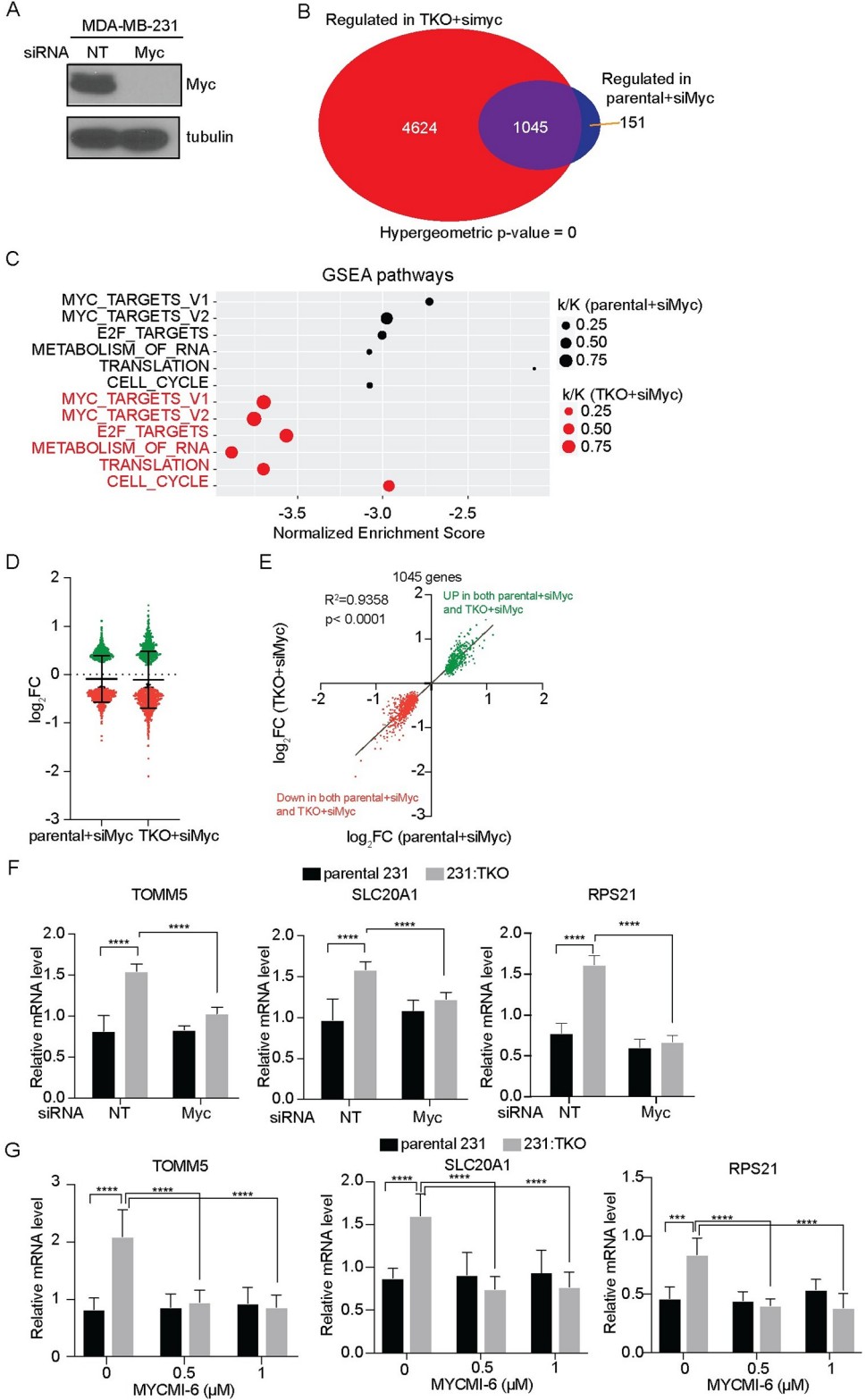

**Fig 6. TXNIP loss expands the Myc transcriptome.** (**A**) Western blotting was used to determine levels of Myc and tubulin in parental 231 cells following a 48-hour treatment with siNT or siMyc. (**B**) Myc-regulated genes in parental 231 and in 231:TKO cells were compared. (**C**) Preranked GSEA was performed using ranked genes lists from Myc-dependent targets in parental 231 and 231:TKO cells and the Hallmark and Reactome data sets from the MSigDB. k/K

value is a ratio of number of genes in our dataset (k) divided by the number of genes in the indicated dataset (K). (**D** and **E**) The magnitude of the Myc dependence of the 1,045 transcripts that are Myc dependent in both parental and 231:TKO cells were compared. (**F** and **G**) The Myc dependence of 3 gene transcripts that were regulated exclusively in 231:TKO cells (Fig 6B) were validated by RT-qPCR following Myc knockdown (**F**) or treatment with the indicated concentrations of the Myc inhibitor MYCMI-6 for 24 hours (**G**). Values are reported as mean and standard deviation. $^{****}p < 0.0001$. The underlying data for Fig 6C–6G can be found in S1 Data. GSEA, Gene Set Enrichment Analysis; MSigDB, Molecular Signatures Database; RT-qPCR, reverse transcription-quantitative PCR; siMyc, siRNA Myc-targeting; siNT, siRNA non-targeting; TKO, TXNIP-knockout; TXNIP, thioredoxin interacting protein.

examining the expression of each of the 1,045 transcripts individually revealed that they were all regulated in the same direction and to a similar degree following Myc knockdown (Fig 6E) in both cell types. These data suggest that TXNIP loss does not fundamentally alter Myc's intrinsic activity as a transcription factor.

There were 4,624 transcripts whose expression showed Myc dependence in 231:TKO cells (Fig 6B) but not in parental 231 cells. We validated the Myc and TXNIP dependence of 3 transcripts, TOMM5, SLC20A1, and RPS21, which were part of this group. Each of these transcripts showed elevated Myc binding following TXNIP loss (Fig 3A). Consistent with our RNA-seq data, the levels of each transcript increased only in 231:TKO cells. Further, this induced expression was decreased following Myc knockdown or by disrupting Myc:Max complexes with MYCMI-6 [44] (Fig 6F and 6G), indicating a Myc dependence. By contrast, reducing Myc levels or inhibiting its transcriptional activity in parental 231 cells did not affect the expression of these transcripts (Fig 6F and 6G). These data support our model that TXNIP loss leads to an expansion of the Myc-dependent transcriptome rather than simply up-regulating Myc-dependent targets that are expressed in parental 231 cells.

## Discussion

We provide evidence that TXNIP loss leads to an up-regulation of Myc-dependent gene signatures. Our data suggest that TXNIP loss does not regulate Myc's intrinsic activity as a transcriptional factor per se, but drives Myc-dependent gene expression by dramatically increasing its binding across the genome. TXNIP loss in 2 TNBC cell lines drove the gene expression programs containing known Myc targets. Further, there is a strong negative enrichment for transcripts correlated with TXNIP expression across breast cancers in the METABRIC database and Myc-associated gene signatures. These findings suggest that TXNIP's ability to regulate Myc's transcription function is not restricted to TNBC cells lines. Furthermore, TXNIP loss in an immortal, but non-transformed, myoblast cell line also resulted in an increase in Myc-associated gene expression programs. Together, these data suggest that TXNIP may be a general regulator of Myc-dependent transcription, capable of functioning in cells from different lineages and transformation status.

Compared to the parental 231 cells, the Myc-dependent transcriptome is expanded in the 231:TKO cells (Fig 7A). This finding suggests that TXNIP loss does not simply increase the expression of the Myc-dependent transcripts present in the parental 231 cells but fundamentally alters Myc-driven transcriptional programs. Our ChIP-seq analysis suggests that the expanded Myc transcriptome in 231:TKO cells results from an increase in global Myc binding (Fig 7B). We show that the increase in Myc binding on G0S2 in 231:TKO cells results in up-regulation of G0S2 expression. G0S2 is one example of target genes where TXNIP loss increases Myc-binding and transcriptional activity, but our analysis suggests that effect of TXNIP loss on Myc activity is global in nature and not restricted to G0S2. Our previous work showed that Myc can repress TXNIP expression by competing with its obligate transcriptional activator MondoA for a double E-box site in its promoter [12]. Collectively, our findings here

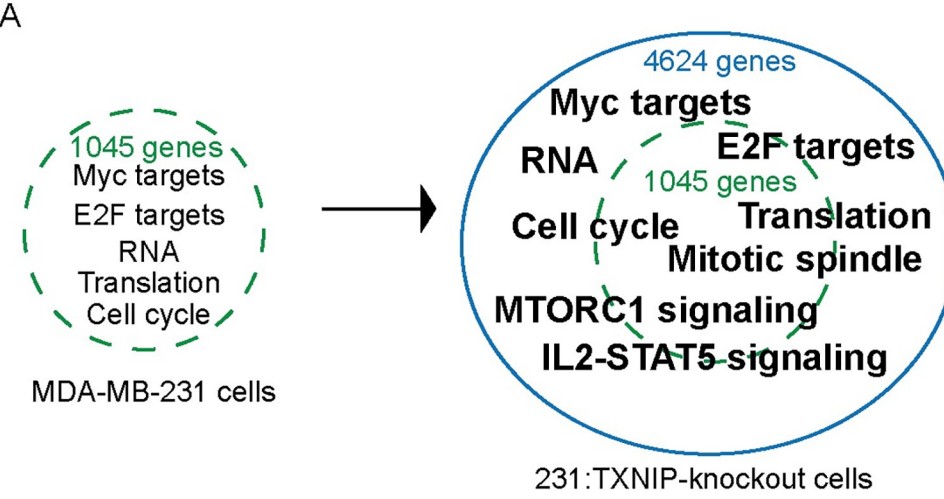

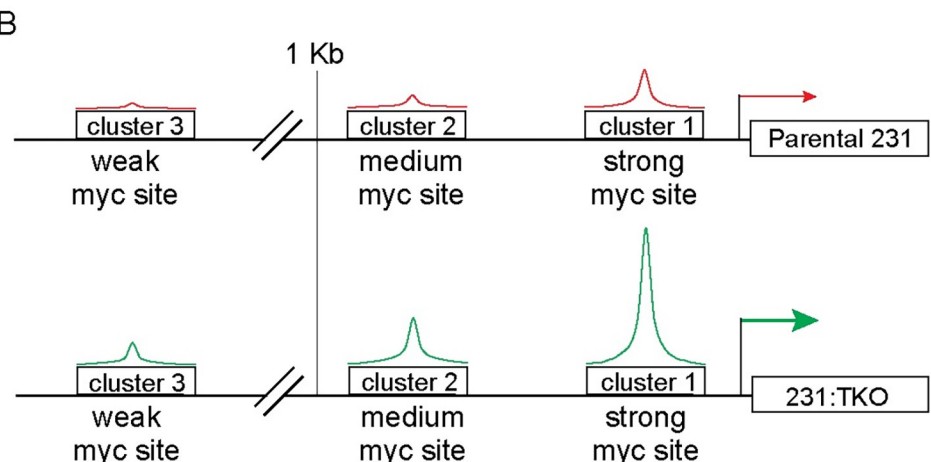

**Fig 7. TXNIP is a repressor of Myc genomic binding and transcriptional activity. (A)** Following TXNIP loss, the Myc-dependent transcriptome is expanded relative to that present in the parental MDA-MB-231 cells. Expansion in the 231:TKO cells does not dramatically alter the pathways regulated in the parental cells; rather, there are more genes in each pathway regulated following loss of TXNIP. (**B**) We assigned Myc binding to 3 clusters based on whether they showed strong (cluster 1), medium (cluster 2), or weak (cluster 3) Myc binding in the parental 231 cells. The majority of sites in clusters 1 and 2 are located within 1 kb of the TSS, whereas the majority of the sites in cluster 3 are more distal from the TSS and likely represent distal enhancers. TKO, TXNIP-knockout; TSS, transcriptional start site; TXNIP, thioredoxin interacting protein.

suggest that Myc-driven repression of TXNIP drives a feedforward regulatory circuit that reinforces Myc transcriptional programs.

Control of TXNIP transcription, translation, and stability is tightly coupled to progrowth signals. For example, mTOR blocks MondoA transcriptional activity, activated Ras blocks translation of TXNIP mRNA, and AKT can phosphorylate TXNIP to trigger its degradation [7,9,45]. One implication of these findings is that progrowth pathways via their suppression of TXNIP expression result in an indirect up-regulation of Myc genomic binding and transcriptional programs. Another consequence of reducing TXNIP expression is an increase in glucose uptake, which we speculate provides carbon backbones for Myc-driven synthesis of macromolecules. This coordination of nutrient availability, i.e., low TXNIP levels and high glucose uptake, with nutrient use, i.e., Myc-driven synthesis of glucose-derived macromolecules, is

likely important for supporting growth and proliferation of many cancer types. Supporting this hypothesis, a $Myc_{high}$/$TXNIP_{low}$ gene signature correlates with poor clinical outcomes in TNBCs, but not in other breast cancer subtypes [12].

Recent studies demonstrate that progressively increasing Myc levels drives Myc to low affinity non-canonical binding sites [27–30] and qualitative changes in Myc-dependent gene expression. The emerging model suggests that at low levels, Myc binds predominately to high affinity sites and regulates expression of genes that carry out essential housekeeping functions. As Myc levels increase, it invades lower affinity sites in promoter and enhancer regulatory regions. From these lower affinity sites, Myc is proposed to drive the expression of genes associated with its function as a transforming oncogene. The expanded Myc-dependent transcriptome in the 231:TKO cells mirrors these findings (Fig 7A and 7B). We divided Myc binding events in 231:TKO cells into high (cluster 1), medium (cluster 2), and low (cluster 3) affinity groups. The highest proportion of high affinity canonical CACGTG Myc binding sites were found in cluster 1, with the proportion decreasing in cluster 3 (Fig 3F). Myc binding events in cluster 3 were farther from the TSS, suggesting that TXNIP loss allows Myc to invade distal enhancer elements. GSEA analysis revealed that high affinity Myc binding events in cluster 1 were associated gene sets enriched for housekeeping functions such as metabolism of RNA. By contrast, the sites in low affinity cluster 3 show enrichment for gene expression programs that may correspond to Myc's function as a transforming oncogene, e.g., Rho signaling. The sites in medium affinity cluster 2 are enriched in gene sets associated with both Myc's housekeeping and transformation-relevant transcriptional targets, suggesting an intermediate phenotype. With the increased level of Myc genomic binding in 231:TKO cells, particularly at potentially transformation-relevant targets, one might expect that they would display a higher level of cell growth-associated phenotypes. This does not appear to be the case at least in normal culture medium replete with glucose and serum-supplied growth factors (S1D Fig). Given this finding, it is somewhat surprising that TXNIP loss almost completely blocked the long-term growth of 231 cells in orthotopic xenograft models (Fig 2G). However, several studies show that high Myc levels or activity can drive apoptosis rather than simply increasing transformation and/or proliferation [46,47]. Further, MondoA is required for Myc-dependent growth/proliferation in different models [48,49]. Given that TXNIP is MondoA's principal target [50], it is possible that TXNIP is also required for Myc-dependent growth/proliferation. Additional experiments are required to distinguish between these possibilities.

In general terms, Myc's function in transcription and transformation is tightly linked to its absolute expression level [25]. TXNIP loss expands Myc genomic binding and drives Myc-dependent gene expression programs, yet Myc protein levels do not increase following TXNIP loss (Figs 1A, S2A and S2D). Myc knockdown experiments showed that its intrinsic activity as a transcriptional activator is similar at Myc-dependent targets expressed in both parental 231 and 231:TKO cells (Fig 6D and 6E). By contrast, Myc genomic binding is dramatically increased in 231:TKO cells compared to parental cells. Together, these data suggest that TXNIP loss drives Myc-associated gene expression programs by increasing Myc's genomic binding rather than by increasing its transcriptional activity. Although, we cannot formally rule out a direct role for TXNIP in regulating Myc genomic binding, we favor a model where TXNIP effects Myc genomic binding indirectly. TXNIP loss increases Myc binding approximately 2-fold at most sites, suggesting an effect on Myc itself or on a common Myc-associated cofactor. Our preliminary experiments provisionally rule out a role for TXNIP in regulating global chromatin accessibility (S9A–S9C Fig), the amount of Myc in the nucleus, the formation of Myc:Max heterodimers, or Myc's association with cofactors required for genome binding such as WDR5 [51] (S10A–S10E Fig). As Myc functions within a complicated network of related transcription factors [52], it is possible that TXNIP loss might reduce the levels of one

or more Myc antagonists, e.g., MXD proteins or Mnt. We do not think this is the case as TXNIP loss does not substantially change the mRNA expression, as revealed by our RNA-seq experiment, of other Myc network members (S10F Fig). Further, a reshuffling of Myc network proteins would be expected to impact the levels of Myc:Max complexes, which we did not observe (S10D and S10E Fig). We have published that Myc and MondoA can compete for a shared double E-Box binding site in the TXNIP promoter [12]; however, our MondoA ChIP-seq data in parental 231 cells suggest that it binds to relatively few Myc sites. This observation suggests that competition between Myc and MondoA for shared binding sites is not common and unlikely accounts for the increase in global Myc genomic binding we see following TXNIP loss. We are currently exploring whether TXNIP regulates Myc:Max genomic occupancy by altering its posttranslational modification state or its association with ancillary factors that stabilize its association with chromatin.

There are 2 general models for Myc-dependent transcriptional activity [53]: (1) Myc functions as a general amplifier of gene expression, affecting the expression of most expressed genes in a given cell or (2) that it functions as a sequence-specific transcriptional activator. Our experiments do not directly address this important question; however, our findings are most consistent with TXNIP loss altering Myc's function as a sequence-specific DNA-binding transcription factor. For example, TXNIP loss led to the up regulation of only 1,792 transcripts, rather than a broad and global up-regulation of gene expression. Further, Myc binding events in the 231:TKO cells, particularly in cluster 1, were enriched for canonical CACGTG binding sites and approximately 51% (Fig 5A) of the Myc-dependent gene expression changes we observed in 231:TKO cells correlated with increased Myc occupancy. Finally, TXNIP loss did not lead to a global opening of chromatin as one might expect if it impacted a putative Myc amplifier function. We also note that TXNIP loss does result in the down-regulation of 742 transcripts, of which about 50% have associated Myc-binding sites. This finding may indicate an enhancement of Myc's transcription repression function, which is typically ascribed to interaction with Miz1/ZBTB17 [54]. However, Miz1/ZBTB17 binding sequences were not enriched in the proximity of Myc-binding sites in the 231:TKO cells. Therefore, while the mechanistic connection remains to be clarified, we favor a model where TXNIP loss drives transcriptional "repression" by an indirect mechanism rather than by increasing interactions between Myc and Miz/ZBTB17.

This study and others demonstrate and establish that TXNIP is a repressor of at least 2 common features of the transformed state: Myc transcriptional activity and glucose uptake [1,5,55,56]. TXNIP expression is exquisitely dependent on the transcription factor MondoA [12,57]. Further, MondoA transcriptional activity seems to be primarily, if not solely, dedicated to regulating TXNIP [50]. Together, these findings suggest that approaches to ectopically activate MondoA transcriptional activity might be a way to block Myc transcriptional activity in therapeutic setting. In this regard, translation initiation inhibitors increase TXNIP expression in a MondoA-dependent manner [58]. TXNIP induction by protein synthesis inhibitors is apparently independent of oncogenic burden, suggesting a broad potential utility of this approach.

## Materials and methods

### Cell culture conditions

Parental MDA-MB-231 and 231:TKO cells were cultured in Dulbecco's Modified Eagle Medium (DMEM) (Gibco; 1195073) with 10% fetal bovine serum (FBS) (Gibco; A3160506), 1X MEM Non-Essential Amino Acids Solution (Gibco; 11140076), and 1X penicillin–streptomycin (Gibco; 15140148). MB135 [33] and MB135:TKO cells were cultured in Ham's F10 with

L-glutamine (Thermo Fisher; 11550043) with 20% FBS (Gibco; A3160506), 1X penicillin–streptomycin (Gibco; 15140148), 10 ng/ml recombinant human Fibroblast Growth Factor (Promega; G5071), and 1 μM dexamethasone (Sigma-Aldrich; D4902). For differentiation, MB135 and MB135:TKO cells were cultured in Ham's F10 with L-glutamine (Thermo Fisher; 11550043), 1X heat-inactivated horse serum (Sigma-Aldrich; H1270), 1X penicillin–streptomycin (Gibco; 15140148), 10 μg/ml insulin from bovine pancreas (Sigma-Aldrich; I-1882), and 10 μg/ml transferrin (Sigma-Aldrich; T-0665). Parental HCC70 and HCC70:TKO cells were cultured in Roswell Park Memorial Institute (RPMI) 1640 Medium (Thermo Fisher Scientific; 11875119) with 10% FBS (Gibco; A3160506), 10 mM N-2-hydroxyethylpiperazine-N-2-ethane sulfonic acid (HEPES) (Thermo Fisher Scientific; 15630080), and 1 mM sodium pyruvate (Invitrogen; 11360070). All cells were maintained at 37°C and 5% $CO_2$.

## CRISPR/Cas9 gene editing

231:TKO cells were generated using human TXNIP CRISPR/Cas9 KO plasmid with TXNIP-specific guide RNA sequences from GeCKO (v2) library (Santa Cruz; sc-400664). TXNIP knockout clones were isolated from single cells, and TXNIP knockout was validated using the polymerase chain reaction (PCR) and western blotting using TXNIP (Abcam) antibodies. MB135:TKO and HCC70:TKO cells were generated using human TXNIP CRISPR/Cas9 KO plasmid, 3 TXNIP-specific guide RNA sequences from GeCKO (v2) library (Santa Cruz; sc-400664), and a homology-directed repair (HDR) construct containing a puromycin resistance cassette (Santa Cruz; sc-400664-HDR). TKO cells were isolated following selection of cells in 2.5 μg/mL puromycin. Loss of TXNIP was verified by immunoblotting.

## Western blotting

Approximately 8 X $10^6$ cells were washed with 1X cold PBS once and scrapped with cell scrapper into ice-cold lysis buffer (400 mM NaCl, 20 mM HEPES [pH7.6], 1 mM EDTA, 1 mM EGTA, 25% glycerol, and 0.1% NP-40) with protease inhibitors (1 mM PMSF, 2.5 μg/ml aprotinin, 1 μg/ml leupeptin, and 1 μg/ml pepstatin), phosphatase inhibitor cocktail 1 (Sigma; P2850), and phosphatase inhibitor cocktail 2 (Sigma; P5726). Cells were disrupted using a bioruptor (Diagenode; UCD-200) with a setting of 15 minutes, 30 seconds on, 30 seconds off. After sonication, disrupted cells were centrifuged at 14,000 rpm for 10 minutes. Supernatants were collected for further analysis. Protein concentrations were determined using a Bio-Rad protein assay (Bio-Rad; 5000006). Equivalent amounts of protein (40 to 80 μg) for different samples were resolved by SDS-PAGE, following transfer to PVDF membrane (Amersham; 10600023) with a setting of 150 V, 400 mA for 1.5 hours at 4°C. After transfer, the PVDF membrane was blocked with 5% non-fat milk in 1X TBST (1X Tris-buffered saline [pH 7.4] with 0.1% Tween-20) for 1 hour. Membranes were probed with primary antibodies using dilution between 1:500 and 1:2,000 (TXNIP, Abcam, ab188865, 1:2,000; c-MYC, Abcam, ab32072, 1:2,000; G0S2, US Biological, 127066, 1:500; and alpha-tubulin, Molecular Probes, 236–10501, 1:20,000) for overnight at 4°C. Protein signals were detected using HRP-conjugated mouse IgG (GE Healthcare, NA931, 1:5,000), HRP-conjugated rabbit IgG (GE Healthcare, NA934, 1:15,000), and ProSignal Pico ECL (Genesee Scientific, 20-300B).

## Reverse transcriptase quantitative PCR (RT-qPCR)

RNA was extracted from cells using Zymo Research Quick RNA Miniprep Kit (Genesee Scientific, 11–328). Approximately 200 ng RNA was used to generate cDNA using GOScript Reverse transcriptase (Promega, A5001). qPCR was performed using CFX Connect Real-Time System and CFX Manager 3.1 program (Bio-Rad). Relative mRNA expression levels were

**Table 1. List of qRT-qPCR primers.**

| Designation | Sequence (5′-3′) | Reference |
|---|---|---|
| c-Myc_forward (human) | TCAAGAGGTGCCACGTCTCC | Shen L et al 2015 [12] |
| c-Myc_reverse (human) | TCTTGGCAGCAGGATAGTCCTT | Shen L et al 2015 [12] |
| TXNIP_forward (human) | TGACTTTGGCCTACAGTGGG | Peterson CW et al 2010 [57] |
| TXNIP_reverse (human) | TTGCGCTTCTCCAGATACTGC | Peterson CW et al 2010 [57] |
| Actin_forward | TCCATCATGAAGTGTGACGT | Peterson CW et al 2010 [57] |
| Actin_reverse | TACTCCTGCTTGCTGATCCAC | Peterson CW et al 2010 [57] |
| TOMM5_forward (human) | CTCCTGCGAGTCACTCCATT | This paper |
| TOMM5_reverse (human) | CTCCTGCGAGTCACTCCATT | This paper |
| SLC20A1_forward (human) | GCAACTCGTGGCTTCGTTTT | This paper |
| SLC20A1_reverse (human) | ACTGGATCTGCCTTATGGAGG | This paper |
| RPS21_forward (human) | TCCGCTAGCAATCGCATCAT | This paper |
| RPS21_reverse (human) | TCATCTGACTCACCCATCCTAC | This paper |
| G0S2_forward (human) | CGAGAGCCCAGAGCCGAGATG | This paper |
| G0S2_reverse (human) | AGCACCACGCCGAAGAG | This paper |

determined from a standard curve generated for each RT primer set. Relative mRNA expression levels for different conditions/samples were normalized to β-actin expression. Three biological replicates of experiments were performed. The values were reported as mean ± standard deviation of 3 technical replicates. Statistical significance was calculated using *t* test. Sequences of the primers used for RT-qPCR are listed in Table 1.

## Myc knockdown and Myc inhibitor treatment

A day before treatment, 0.5 X $10^6$ parental 231 and 231:TKO cells were plated in a 6-well plate. For Myc knockdown assay, cells were treated either 25 nM siRNA non-targeting (siNT) (Dharmacon; D-001810-10-05) or siRNA Myc-targeting (siMyc) (Dharmacon; L-003282-02-0005) for 48 hours. For Myc inhibitor treatment, cells were treated with different concentrations of Myc inhibitor, MYCMI-6 (MedChemExpress; 681282-09-7) or DMSO for 24 hours.

## Analysis of METABRIC data

The gene expression data for 1,904 breast cancer samples was obtained from cBioPortal [59,60] as part of the METABRIC dataset [34] and reported as z-scores relative all samples. The stats package in R was used to calculate Pearson coefficients for every measured gene correlated to TXNIP. Genes were then ordered by their Pearson correlation coefficients and used as a preranked dataset for GSEA analysis.

## Liquid chromatography–mass spectrophotometry (LC–MS)

Approximately 20 X $10^6$ parental 231 cells and 231:TKO cells were cultured in DMEM (Gibco; 1195073) with 10% FBS (Gibco; A3160506), 1X MEM Non-Essential Amino Acids Solution (Gibco; 11140076), and 1X penicillin–streptomycin (Gibco; 15140148). 231:TKO cells were treated either 25 nM siNT (Dharmacon; D-001810-10-05) or siMyc (Dharmacon; L-003282-02-0005) for 48 hours. Cells were trypsinized and centrifuged at 1,500 rpm for 5 minutes. Cell pellets were resuspended with 1X PBS. Resuspended cells were distributed between two 1.5 mL cryovials (approximately 9 x $10^6$ cells each), which were then centrifuged at 1,500 rpm for 5 minutes. The supernatant was aspirated and dried cell pellets were immediately flash frozen in liquid nitrogen and stored in −80˚C until ready to proceed to LC–MS. Briefly, to prepare

samples for LC–MS, each cell pellet was extracted with 100 μL of ice-cold 3:2 acetonitrile (ACN):double distilled $H_2O$ (dd$H_2O$ and 0.1% ammonium hydroxide containing 0.1 μg/mL carnitine-d9 while on dry ice. Samples were vortexed for 30 seconds and sonicated for 1 minute on ice. Samples were then chilled at −20°C for 1 hour, followed by centrifugation at 20,000 rcf at 4°C for 10 minutes. Samples were combined into new Eppendorf tubes and the precipitate discarded. A Sequant ZIC-pHILIC 2.1 x 100 mm column (Millipore Sigma) with Phenomenex Krudkatcher (Phenomenex, Torrence, CA, USA) was used for chromatographic separation. Buffer A (95% dd$H_2O$, 5% ACN, and 25 mM ammonium carbonate) and buffer B (95% ACN, and 5% dd$H_2O$) were used for chromatographic separation. An initial concentration of 95% buffer B and 5% buffer A was held for 3.3 minutes at a flow rate of 0.15. Buffer B was decreased to 15% over 7.5 minutes and held for 3.8 minutes. Mass spectral analysis was optimized and performed on a SCIEX 6500 QTRAP. Data were analyzed using SCIEX Multi-Quant software. The identities of metabolites were determined by METLIN [61]. Data were normalized and analyzed using MetaboAnalyst [62]. Log transformation, Pareto scaling, and quantile transformation were used for sample normalization. Enriched metabolites were plotted using Prism. Pathway analysis was performed on enriched metabolites using Functional Analysis module in MetaboAnalyst [63].

## Chromatin immunoprecipitation sequencing (ChIP-seq)

Approximately 20 X $10^6$ MDA-MB-231 or 231:TKO cells were cross-linked with 1% formaldehyde for 10 minutes at room temperature and then were treated with 0.125 M glycine for 5 minutes to quench the cross-linking reaction. Cross-linked cells were washed with 1X cold PBS and then were lysed in Farnham lysis buffer (5 mM PIPES [pH 8.0], 85 mM KCl, 0.5% NP-40). Fixed chromatin was then harvested in Farnham lysis buffer supplemented in protease and phosphatase inhibitors (Thermo Fisher Scientific; A32959) by scraping the plates with cells scrapers and transferring the material to 15 mL conical tubes. Fixed chromatin was centrifuged at 2,000 rpm for 5 minutes at 4°C, and pellets were resuspended in RIPA buffer (1X PBS, 1% NP-40, 0.5% sodium deoxycholate, 0.1% SDS) for sonication. Sonication was performed on an Active Motif EpiShear Probe Sonicator for 5-minute cycles of 30 seconds on and 30 seconds off with a 40% amplitude. Sonicated samples were centrifuged at 14,000 rpm for 15 minutes at 4°C, and supernatants were collected. Sonicated DNA with 200 to 500 base pair fragments was used for immunoprecipitation. For immunoprecipitation, 5 μg Myc (Cell Signaling; D3N8F) antibody was used. Anti-Myc was bound to Dynabeads M-280 sheep anti-rabbit (Invitrogen; 11204D) for at least 2 hours at 4°C. Bead-antibody complexes were incubated with fragmented chromatin overnight at 4°C with rocking. After overnight incubation, Dynabeads containing antibody-bound chromatin were captured with a magnetic rack. Dynabeads were washed with LiCl wash buffer (100 mM Tris [pH 7.5], 500 mM LiCl, 1% NP-40, and 1% sodium deoxycholate) for 5 times at 4°C. Each wash was 3 minutes with rocking. After washing, the Dynabeads were washed once with 1 ml TE buffer (10 mM Tris–HCl [pH 7.5] and 0.1 mM $Na_2$EDTA). Beads were resuspended in 200 ml IP elution buffer (1% SDS and 0.1 M NaHCO$_3$) with vortexing. Beads were incubated in a 65°C bead bath for 1 hour with vortexing the tubes every 15 minutes to elute the antibody-bound chromatin from the beads. Beads were centrifuged at 14,000 rpm for 3 minutes at room temperature. Supernatants (immune-bound chromatin) were collected and transferred to new microcentrifuge tubes. Input samples for each condition served as controls. Immuno-bound chromatin and inputs were de-cross-linked in a 65°C bead bath for overnight.

Reverse cross-linked DNA was purified with ChIP DNA Clean & Concentrator Kit (Zymo Research; 11379C) according to the manufacturer's protocol. Eluted DNA in EB buffer was

used to construct the ChIP library. Preparation of immunoprecipitated DNA for sequencing was performed as previously described [63]. Briefly, blunted DNA fragments were ligated with sequencing adapters. The ligated DNA fragments were amplified with library PCR primers that contain barcodes (NEBNext ChIP-Seq Library Prep Reagent for Ilumina) for 15 cycles. Amplified DNA libraries from the anti-Myc ChIP were sequenced using Illumina HiSeq Sequencing with 50 cycles of single read. The resulting Fastq files were aligned to the human genome (hg19) using NovoAlign. Peaks were called using Model-Based Analysis of ChIP-seq-2 (MACS2) [64] using a $p$-value cutoff of 0.01 and the mfold parameter between 5 and 50. The heatmap was generated using deepTools 2.0 [65]. MYC-bound genes were annotated using the R package ChIPseeker [36]. Myc binding motifs were determined using HOMER [37]. GSEA and preranked GSEA [31,32] were used to determine the pathway enrichment of Myc-bound genes. ggplot2 [66] was used to draw dot plots for pathway enrichment.

## Chromatin immunoprecipitation quantitative PCR (ChIP-qPCR)

Anti-Myc immunoprecipitations were performed from chromatin isolated from $20 \times 10^6$ MDA-MB-231 or 231:TKO cells as described above. Myc binding levels on genes were assessed with qPCR. qPCR was performed as described above. ChIP-qPCR primers that were used for experiments: human G0S2, forward:5′-TTCGCGTGCACACTGGCCTTCCC-3′, reverse: 5′-GAGGAGGGGAAAAGGAGGGGGTGGAAC-3′.

## RNA sequencing library construction and analysis

Total RNA was extracted from $3 \times 10^6$ cells using the Zymo Research Quick RNA miniprep kit (Zymo Research; R1055). Approximately 500 ng of total RNA was used to capture mRNA and construct libraries using KAPA Stranded mRNA-Seq Kit (KAPA Biosystems; KK8420) according to the manufacturer's protocol. Briefly, after mRNA capture and fragmentation, cDNA was synthesized. Sequencing adaptors were ligated to the cDNA fragments. The adaptor-ligated cDNA fragments were amplified with library PCR primers that contain barcodes for about 10 cycles. Libraries were sequenced using either Illumina HiSeq 50 cycles Single-Read Sequencing or Novaseq Paired-Read Sequencing. The resulting Fastq files were aligned to the human genome (hg38) using STAR [67]. Counts were generated using FeatureCounts version 1.63 [68] with the arguments "-T 24 -p -s 2 –largestOverlap" and using the Ensemble Transcriptome build 102 for GRCh38. DESeq2 [69] was used to determine the differential gene expression in different samples or treatments. Raw counts, rlog values, and normalized counts for each sample or treatment were generated by the DESeq2 program. Counts ≤5 were filtered and an adjusted $p$-value less than or equal to 0.05 was used in the DESeq2 program to determine differential gene expression. GSEA and preranked GSEA [31,32] were used to determine the pathway enrichment of Myc-bound genes. ggplot2 [66] was used to draw dot plots for pathway enrichment.

## Assay for transposase-accessible chromatin with high-throughput sequencing (ATAC-seq)

ATAC-seq was performed on parental MDA-MB-231 and 231:TKO cells with 250,000 cells for each library as described [70]. Briefly, 250,000 cells were trypsinized and pelleted, washed once with cold 1X PBS, resuspended, and lysed in 50 μL of ATAC-lysis buffer (ATAC resuspension buffer: 10 mM Tris–HCl [pH 7.4], 3 mM MgCl$_2$, 10 mM NaCl containing 0.1% NP40, 0.1% Tween-20, and 0.1% Digitonin), and incubated for 3 to 4 minutes on ice. Lysis was quenched with 1 mL of wash buffer (ATAC resuspension buffer containing 0.10% Tween-20); lysed nuclei were counted using Countess II cell counter. Nuclei were pelleted and resuspended in

transposition mixture using an in-house prepared Tn5 transposase as described [71]. Sequencing reads were aligned to hg19 using Bowtie [72] with the following parameters: -m 1 -t–best -q -S -l 32 -e 80 -n 2. SAM files were converted to BAM files and sorted using samtools [73]. Peaks were called using MACS2 [64] using a *p*-value cutoff of 0.01 and the mfold parameter between 5 and 50. The heatmap was generated using deepTools 2.0 [65].

### Proliferation assays

A total of 6,000 cells per well were plated in 96-well plates. For the assays with Myc knockdown, 0.5 X $10^6$ cells per well of a 6-well plate were used. Cells were treated with either 25 nM siNT (Dharmacon; D-001810-10-05) or siMyc (Dharmacon; L-003282-02-0005), 24 hours after plating. Proliferation was monitored for 48 hours after siRNA reagents were added. Proliferation was monitored for 2 days on the Incucyte Zoom Live Cell Imaging Platform (Sartorius) with 10X magnification, and images were captured at 2-hour intervals.

### Orthotopic xenograft experiments

Cell lines were tested for mycoplasma contamination prior transplantation. Approximately 100,000 of either parental 231 or 231:TKO cells were transplanted into the cleared mammary fat pads of 3- to 5-week-old immunocompromised NOD-scid (NOD.CB17-Prkcd/J) mice as described and under the guidance of the Preclinical Research Resource at HCI [74]. Ten mice were used for each cell line. Starting 7 days after the transplantation surgery, tumors were measured weekly using calipers. The experiment was terminated after 64 days. A representative of 2 independent experiments is shown. All animal experiments were evaluated and approved by the University of Utah's Institutional Animal Care and Use Committee under protocol number 15–04012.

### Supporting information

**S1 Fig. TXNIP loss mimics Myc overexpression. Related to Fig 1**. (**A**) The relative *TXNIP* mRNA levels (normalized to that of β-actin) in parental 231 and 231:TKO cells were determined by RT-qPCR. (**B**) The expression levels of Myc and TXNIP from approximately 70 breast cancer cells were plotted. Their expression in HCC70 and MDA-MB-231 cells is indicated. (**C**) Western blotting was used to determine the levels of Myc, TXNIP, and tubulin in 3 biological replicates of parental 231 and 231:TKO cells. (**D**) Cell proliferation for parental 231 and 231:TKO cells in regular medium over a 48-hour time course was measured based on the percentage of confluency using real-time videography. (**E**) A volcano plot of the fold changes and adjusted *p*-values of regulated transcripts in 231:TKO cells. Gene expression changes in 231:TKO cells were determined using DESeq2. (**F**) A preranked GSEA enrichment plots of the regulated transcripts in 231:TKO cells with the indicated Hallmark datasets. The underlying data for S1A, S1B, S1D and S1E Fig can be found in S1 Data. GSEA, Gene Set Enrichment Analysis; RT-qPCR, reverse transcription-quantitative PCR; TKO, TXNIP-knockout; TXNIP, thioredoxin interacting protein.
(TIF)

**S2 Fig. TXNIP loss mimics Myc overexpression is not restricted to breast cancer cell lines. Related to Fig 1**. (**A**) Western blotting was used to determine the levels of Myc, TXNIP, and tubulin in 3 biological replicates of parental HCC70 and HCC70:TKO cells. (**B**) The levels of Myc protein from (**A**) in parental HCC70 and HCC70:TKO were quantified using ImageJ. (**C**) Cell proliferation for parental HCC70 and HCC70:TKO cells in regular medium over a 48-hour time course was measured based on the percentage of confluency using real-time videography. (**D**) Western blotting was used to determine the levels of Myc, TXNIP, and tubulin in parental MB135 and MB135:TKO cells. (**E**) The levels of Myc protein from (**D**) in parental

MB135 and 135:TKO were quantified using ImageJ. (**F**) Cell proliferation for parental MB135 and MB135:TKO cells in regular medium over a 48-hour time course was measured based on the percentage of confluency using real-time videography. (**G**) A preranked GSEA enrichment plot of regulated transcripts in differentiated myoblast MB135:TKO cells with the Hallmark_-Myc_Targets_V1 dataset. The underlying data for S2B, S2C, S2E and S2F Fig can be found in S1 Data. GSEA, Gene Set Enrichment Analysis; TKO, TXNIP-knockout; TXNIP, thioredoxin interacting protein.
(TIF)

**S3 Fig. TXNIP-correlated gene expression programs are negatively correlated with pro-growth pathways. Related to Fig 1.** (**A**) Transcripts positively correlated TXNIP expression across almost 2,000 breast tumors are negatively correlated with genes in the 4 shown Hallmark datasets or (**B**) positively correlated with genes in the 4 shown Hallmark datasets.
(TIF)

**S4 Fig. Gene expression changes in 231:TKO cells are Myc dependent. Related to Fig 2.** (**A**) Cell proliferation for 231:TKO+siNT and 231:TKO+siMyc cells in regular medium over a 48-hour time course was measured based on the percentage of confluency using real-time videography. (**B**) A preranked GSEA was preformed using a ranked list of the differentially expressed genes in 231:TKO+siMyc cells and the indicated Hallmark and Reactome datasets. (**C**) A ranked list of the 548 reciprocally regulated genes in Fig 2D were used in a GSEA using the MSigDB and the Hallmark and Reactome datasets. k/K value is a ratio of number of genes in our dataset (k) divided by the number of genes in the indicated dataset (K). (**D**) Expression changes of 24 ribosomal protein genes regulated 231:TKO and 231:TKO+siMyc datasets. The underlying data for S4A, S4C and S4D Fig can be found in S1 Data. GSEA, Gene Set Enrichment Analysis; MSigDB, Molecular Signatures Database; siMyc, siRNA Myc-targeting; siNT, siRNA non-targeting; TKO, TXNIP-knockout.
(TIF)

**S5 Fig. G0S2 is reciprocally regulated by TXNIP and Myc. Related to Fig 4.** (**A**) Genome browser view from RNA sequencing of human *G0S2* (G0S2) mRNA in parental 231 and 231:TKO cells. (**B**) Myc binding, as visualized using IGV_2.5.2, to G0S2 in parental 231 and 231:TKO cells. Putative double E-Box element in the G0S2 promoter [41] encompassed the Myc-binding site. (**C**) Luciferase activities of rat G0S2 reporter in parental and 231:TKO cells with ectopic expression of human TXNIP from pcDNA3 vector or pcDNA3 EV were measured. Luciferase activity was normalized to β-gal activity. At least 2 biological replicates were carried out for all luciferase experiments. Representative figures were shown. Values are reported as mean and standard deviation. $^{**}p < 0.01$; $^{***}p < 0.001$. (**D** and **E**) The luciferase activities from WT rat G0S2-luciferase reporter construct and mut rat G0S2-luciferase reporter construct in lysates from parental 231 (**D**) and 231:TKO (**E**) were measured. The double E-Box element of G0S2 promoter in the rat G0S2-luciferase construct was deleted using site-directed mutagenesis [41]. At least 2 biological replicates were carried out for all luciferase experiments. Representative figures were shown. Values are reported as mean and standard deviation. $^{**}p < 0.01$; $^{***}p < 0.001$. The underlying data for S5C–S5E Fig can be found in S1 Data. EV, empty vector; mut, mutated; TKO, TXNIP-knockout; TXNIP, thioredoxin interacting protein; WT, wild-type; β-gal, beta-galactosidase.
(TIF)

**S6 Fig. TXNIP loss drives Myc-dependent gene expression programs. Related to Fig 5.** (**A**) The list of 2,903 genes that showed increased Myc binding in 231:TKO cells compared to parental 231 cells were ranked according to their differential expression in 231:TKO+siMyc

cells. This list was analyzed using preranked GSEA to identify enriched pathways in the MSigDB. k/K value is a ratio of number of genes from our dataset (k) divided by the number of genes in the indicated dataset (K). (**B**) Heatmap of genes regulated in 231:TKO cells that are enriched in the Reactome oxidative phosphorylation dataset. Differential regulation in 231: TKO+siMyc cells are indicated by yellow (up-regulation) or green (down-regulation) boxes. The distances of Myc binding sites from TSS are determined using GREAT [43]. The genes that have a Myc binding event within 1 kb or more than 1 kb from TSS are indicated in brown. Open boxes indicate no Myc binding. (**C** and **D**) Differentially down-regulated (**C**) or up-regulated genes (**D**) in 231:TKO+siMyc cells were divided into groups based on the number Myc sites associated with each gene. (**E** and **F**) The distribution of Myc binding loci relative to the TSS for down-regulated (Myc-activated targets) (**E**) and up-regulated (Myc-repressed targets) (**F**) genes in 231:TKO+siMyc cells were annotated using ChIPseeker. The distance to the TSS was then compared change in gene expression following Myc knockdown. The underlying data for S6A, S6C and S6D Fig can be found in S1 Data. GREAT, Genomic Regions Enrichment of Annotations Tool; GSEA, Gene Set Enrichment Analysis; MSigDB, Molecular Signatures Database; siMyc, siRNA Myc-targeting; TKO, TXNIP-knockout; TSS, transcriptional start site; TXNIP, thioredoxin interacting protein.
(TIF)

**S7 Fig. Myc-dependent and TXNIP-regulated genes in 231:TKO cells are conserved. Related to Fig 5**. (**A**) The Myc binding sites in our Myc ChIP-sequencing in 231:TKO dataset was compared with Myc binding sites in published Myc ChIP-sequencing datasets GSE77356 and GSE126207 [27,42]. The narrow peaks of each dataset were compared using bedtools intersect to identify overlapped peaks in the 3 datasets [75]. The Venn diagram was drawn using a VennDiagram package in R studio. (**B**) The distance of the 11,145 overlapped Myc binding sites from TSS was annotated using the ChIPseeker program. (**C**) Myc binding motifs were determined using HOMER. (**D**) A total of 11,145 overlapped narrow Myc-binding peaks were annotated with genes using GREAT [43]. A total of 7,871 genes that are annotated from 11,145 overlapped narrow peaks were compared with regulated genes in 231:TKO+siMyc cells. The Venn diagram was drawn using a VennDiagram package in R studio. (**E**) Preranked GSEA using a ranked list of 3,070 Myc-dependent and Myc-bound genes and the Hallmark and Reactome datasets in the MSigDB. k/K value is a ratio of number of genes in our dataset (k) divided by the number of genes in the indicated dataset (K). ggplot2 was used to draw dot plots for pathway enrichment. The underlying data for S7E Fig can be found in S1 Data. GREAT, Genomic Regions Enrichment of Annotations Tool; GSEA, Gene Set Enrichment Analysis; HOMER, Hypergeometric Optimization of Motif EnRichment; MSigDB, Molecular Signatures Database; siMyc, siRNA Myc-targeting; TKO, TXNIP-knockout; TSS, transcriptional start site; TXNIP, thioredoxin interacting protein.
(TIF)

**S8 Fig. TXNIP loss expands the Myc transcriptome. Related to Fig 6**. (**A**) The fold decrease in Myc mRNA by siMyc in both parental 231 and 231:TKO cells was extracted from our RNA sequencing data. (**B**) Cell proliferation for parental 231 with siNT or siMyc in regular medium over a 48-hour time course was measured based on the percentage of confluency using real-time videography. (**C**) The differential expression of genes in 3 different functional groups in parental 231+siMyc and 231:TKO+siMyc datasets were compared. The underlying data for S8B and S8C Fig can be found in S1 Data. siMyc, siRNA Myc-targeting; siNT, siRNA non-targeting; TKO, TXNIP-knockout; TXNIP, thioredoxin interacting protein.
(TIF)

**S9 Fig. TXNIP loss does not increase chromatin accessibility.** (**A** and **B**) To determine the accessibility of chromatin, we used to perform ATAC-seq in both parental 231 and 231:TKO cells. Heatmaps showing the chromatin accessibility in parental 231 and 231:TKO across the entire genome (**A**) and at Myc-binding sites (**B**). Heatmaps were generated within 2 kb upstream and downstream of the accessibility peaks using a *p*-value cutoff of 0.01. (**C**) Genome browser views of chromatin accessibility in the proximity of RPL10A and SLC18B1 in parental 231 and 231:TKO cells. ATAC-seq, assay for transposase-accessible chromatin using sequencing; RPL10A, ribosomal protein L10a; SLC18B1, solute carrier family 18 member B1; TKO, TXNIP-knockout; TXNIP, thioredoxin interacting protein.
(TIF)

**S10 Fig. TXNIP loss does not change activity of Myc:Max complexes.** (**A** and **B**) Western blotting was used to determine the levels of indicated proteins in whole cell lysates (**A**), cytoplasmic and nuclear fractions (**B**) from parental 231 and 231:TKO cells. (**C**) Myc protein was immunoprecipitated with c-Myc antibodies from rabbit (rab) and mouse, respectively. Associated proteins were detected by immunoblotting. (**D**) Max protein was co-immunoprecipitated with c-Myc antibodies from the cytoplasmic (cyto) and nuclear fractions of parental 231 and 231:TKO cells. The levels of immunoprecipitated Max in these cell lines were assessed by western blotting. (**E**) Max protein was immunoprecipitated with Max antibody. Associated proteins were detected by immunoblotting. A Mlx IP in was used as a specificity control. The indicated immunoprecipitated proteins were examined by western blotting. (**F**) Expression levels of the members of the extended Myc family of transcription factors expressed as normalized counts from our RNA-seq analysis. IP, immunoprecipitation; TKO, TXNIP-knockout; TXNIP, thioredoxin interacting protein.
(TIF)

**S1 Data. Underlying data for Figs 1B, 1C, 2B, 2D, 2E, 2F, 2G, 2H, 3C, 3D, 4A, 4C, 4D, 4F, 4G, 5D, 6C, 6D, 6E, 6F, 6G, S1A, S1B, S1D, S1E, S2B, S2C, S2E, S2F, S4A, S4C, S4D, S5C, S5D, S5E, S6A, S6C, S6D, S7E, S8B and S8C.**
(XLSX)

**S1 Raw Images. Uncropped images for blots used in Figs 1A, 2A, 4B, 4E, 6A, S1C, S2A, S2D, S10A, S10B, S10C, S10D and S10E.**
(PDF)

## Acknowledgments

We thank members of the Ayer, Gertz, and Varley labs for their thoughtful comments throughout this project and Elizabeth Leibold for comments on the manuscript.

## Author Contributions

**Conceptualization:** Tian-Yeh Lim, Blake R. Wilde, Kristin E. Murphy, Katherine E. Varley, Jason Gertz, Donald E. Ayer.

**Formal analysis:** Tian-Yeh Lim, Blake R. Wilde, Mallory L. Thomas, Kristin E. Murphy, Jeffery M. Vahrenkamp, Megan E. Conway, Donald E. Ayer.

**Funding acquisition:** Katherine E. Varley, Jason Gertz, Donald E. Ayer.

**Investigation:** Tian-Yeh Lim, Blake R. Wilde, Mallory L. Thomas, Kristin E. Murphy, Megan E. Conway.

**Methodology:** Tian-Yeh Lim, Jason Gertz.

**Project administration:** Donald E. Ayer.

**Resources:** Donald E. Ayer.

**Supervision:** Katherine E. Varley, Jason Gertz, Donald E. Ayer.

**Writing – original draft:** Tian-Yeh Lim, Donald E. Ayer.

**Writing – review & editing:** Tian-Yeh Lim, Donald E. Ayer.

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
