## [Editor Report · Decision Letter 0]

1 Aug 2022

Dear Dr Ayer, 

Thank you for submitting your manuscript entitled "TXNIP loss expands Myc-dependent transcriptional programs by increasing Myc genomic binding" for consideration as a Research Article by PLOS Biology. Please accept my sincere apologies for the long delay in getting back to you as we consulted with an academic editor about your submission.

Your manuscript has now been evaluated by the PLOS Biology editorial staff, as well as by an academic editor with relevant expertise, and I am writing to let you know that we would like to send your submission out for external peer review.

Once your full submission is complete, your paper will undergo a series of checks in preparation for peer review. After your manuscript has passed the checks it will be sent out for review. To provide the metadata for your submission, please Login to Editorial Manager (https://www.editorialmanager.com/pbiology) within two working days, i.e. by Aug 03 2022 11:59PM.

Kind regards,

Richard

Richard Hodge, PhD

Associate Editor, PLOS Biology

rhodge@plos.org

PLOS

---

## [Decision Letter · Decision Letter 1]

9 Sep 2022

Dear Dr Ayer,

Thank you for your patience while your manuscript "TXNIP loss expands Myc-dependent transcriptional programs by increasing Myc genomic binding" was peer-reviewed at PLOS Biology. Please accept my sincere apologies for the delays that you have experienced during the peer review process. Your manuscript has been evaluated by the PLOS Biology editors, an Academic Editor with relevant expertise, and by three independent reviewers.

The reviews are attached below. You will see that the reviewers find your manuscript interesting and well done, but raise overlapping concerns with the overall strength of the mechanistic insights into how TXNIP regulates Myc transcriptional activity. In addition, they note that the physiological and biological significance of the findings should be strengthened, by including data from several cell lines and using in vivo mouse models to demonstrate a role for TXNIP in tumorigenesis.

Based on the reviews and following discussion with the academic editor, I regret that we cannot accept the current version of the manuscript for publication. We remain interested in your study and would be willing to consider resubmission of a comprehensively revised version that thoroughly addresses all the reviewers' and Academic Editor’s comments. We cannot make any decision about publication until we have seen the revised manuscript and your response to the reviewers' comments. Your revised manuscript would be sent for further evaluation by the reviewers.

We appreciate that these requests represent a great deal of extra work, and we are willing to relax our standard revision time to allow you 6 months to revise your study. Please email us (plosbiology@plos.org) if you have any questions or concerns, or envision needing a (short) extension.

**IMPORTANT - SUBMITTING YOUR REVISION**

*Resubmission Checklist*

*Published Peer Review*

*PLOS Data Policy*

*Blot and Gel Data Policy*

Sincerely,

Richard

Richard Hodge, PhD

Associate Editor, PLOS Biology

rhodge@plos.org

REVIEWS:

Reviewer #1: In this paper, Lim et al have examined the consequences of TXNIP gene knockout in the MDA-MB231 TNBC cell line. They found nearly 1800 differentially expressed transcripts to be altered in response to this loss. Interestingly, among the most highly and selectively enriched gene sets were those which are known direct Myc targets. The change in Myc target expression in response to RXNIP inactivation is both quantitative and qualitative; previously bound target genes increase their Myc binding by an average of two-fold and genes that did not bind Myc previously in control cells now bind Myc in TXNIP KO cells. The other top gene sets encoded proteins involved in mitochondrial structure and function, translation and cell cycle. It is known that many of the genes within these sets are also direct Myc target. Overall, these findings indicate TXNIP to be a generalize repressor of Myc target gene regulation.

MAJOR POINTS:

1. The authors should state precisely how many known Myc targets are enriched (both positively and negatively) 

2. The gene sets described in Fig. 1B other than "Myc Target Genes" represent many functions that nonetheless have been described as being Myc-regulated (i.e. mitochondrial structure/function, translation, cell cycle). How many of the genes within these sets are known to be direct Myc targets? Knowing this would allow the authors to increase the number of genes that they state are Myc targets but are not listed as such simply because they are classified as being members of another category. Another way to address the point above is to subtract from the 1800 differentially expressed 231-TKO genes those that are identified as Myc targets and then ask whether the remainder have ever been identified as being direct Myc targets in any other cell lines by searching the ENCODE data base. The Myc target genes identified by the authors represents only a tissue-specific snapshot of potential targets. Doing a broader survey of cell lines other than MB231 might be a more sensitive and comprehensive way of identifying those TXNIP targets that are either definite direct Myc targets or have the potential to be as a result of TXNIP KO. 

3. The authors should more thoroughly discuss the fact that the numbers of both positive and negative Myc targets seem to increase in response to TXNIP inactivation. The former are those whose activity is due to Myc's binding directly to the 5'ends of genes via either E boxes or ChoREs (well discussed in the ms.) whereas the latter suppress transcription, most likely indirectly by binding to and interfering with transcription factors such as Miz1 and Sp1 that bind to inR elements and Sp1 sites, respectively (not as well-discussed). Given that both types of interactions seem to be affected by TXNIP KO, it would seem logical to attribute this to some fundamental change in either Myc or Max. The most obvious type of change would be one that affected Myc-Max heterodimerization or binding to E-boxes/ChoRES and to Miz1/Sp1. This could happen if the fraction of Myc that associated with Max was increased, either via more efficie Myc-Max association or by decreasing the abundance of one of the Mxd members that competes with Myc for Max occupancy. The former state might be the result of some type of post-translational modification of Myc and/or Max that increased their ability to heterodimerize or bind DNA afterwards. This could be due to an altered intracellular redox state, which seems highly likely given that TXNIP plays a major role in maintaining redox balance via its inhibitory effect on thioredoxin. Although the authors' observation is a fascinating one, it lacks any mechanistic insight into how this is occurring. At the very least, the authors should investigate the relative abundance of Myc-Max and relevant Mxd-Max heterodimers. Reduced levels of one or more Mxd members might also drive a greater amount of Myc-Max heterodimer formation simply via mass action. As to determining the underlying basis by which Myc family heterodimer levels might be altered, the authors should consider investigating the redox state of their MDA-MB231 control and TXNIP KO cells. A very straight forward and easy way to do this would be to employ roGFP, which can be targeted to either the cytoplasm or mitochondria (Hanson et al. J Biol Chem. 2004 Mar 26;279(13):13044-53). 

MINOR POINTS: 

Fig. 1E shows a FDR q value of <0 which doesn't make sense. The actual p value should be indicated. In addition, for C and D, both P values are >0.05. The authors should state in the figure legend the criteria used that allow them to justify representing these q values as indicating significant enrichment. 

Fig. 3. Are the 3 clusters distinguished in anything other than the distance their Myc binding sites are from the TSS of their respective genes and the affinity of these sites for Myc? From these data, is it fair to say that, as a rule of thumb, low-affinity Myc binding sites tend to be located progressively further away from the TSS?

Fig. 3F & line 270. The authors should explain to the more general reader what the k/K ratio represents.

Line 330-332. The authors refer to a ChORE element in the rat GOS2 gene that is just upstream of the TSS and that is presumably the functional equivalent of the somewhat degenerative element they identified in the human GOS2 gene's promoter (lines 307-308). How related/conserved are these two elements? 

Line 377: The title of this section is: "TXNIP Controls Myc Transcriptional Programs by increasing Myc Binding". This should be changed given that it is the LOSS of TXNIP that increases Myc binding (lines 396-397: "These results suggest that TXNIP loss increases Myc-dependent gene expression by increasing Myc genome occupancy.") 

Reviewer #2: In "TXNIP loss expands Myc-dependent transcriptional programs by increasing Myc genomic binding", Lim and colleagues explore the role of TXNIP in negatively regulating MYC DNA binding in triple-negative breast cancer. The Authors begin by showing in a TNBC line that knockdown of TXNIP leads to upregulation of transcriptional programs associated with MYC, and this pattern is recapitulated in human breast cancer samples and an unrelated muscle cell line. siRNA-mediated knockdown of MYC in the background of TXNIP loss further solidifies this association. The Authors next used ChIP-sequencing to look at how MYC DNA binding changes in the absence of TXNIP. Interesting, they show three clusters of MYC binding; the strongest cluster bound to classic MYC proximal promoter sites, while the weakest cluster bound to more distal and less class MYC sites. Finally, the Authors knock down MYC in a TXNIP-competent cell line and show that while the number of MYC sites in the presence or absence of TXNIP varies, the average degree to which MYC upregulates its targets does not vary. Overall, this suggests that TXNIP, through a still-unknown mechanism, impacts MYC genomic binding availability but not MYC activity.

 This manuscript generally accomplishes what it sets out to determine, which is the role of TXNIP in restraining MYC genomic binding and activity in TNBC. The Authors successfully backed up their RNA-seq and ChIP-seq data with specific examples throughout the paper, and the finding in Figure 6 that enhanced MYC DNA binding does not necessarily lead to higher levels of gene expression is particularly interesting and important. There are a few issues that will need to be addressed, detailed in the list below. Most important of these are comparing different knockdown efficiencies of MYC in their two MDA-MB-231 lines, including proliferation assays for other manipulations besides TXNIP knockdown, and potentially expanding on the mechanistic role of TXNIP in controlling MYC activity, which is hinted at in the Discussion but without data. 

Major (essential) points:

* The comparison between 231:TKO siMYC and 231 parental siMYC is complicated by different knockdown efficiencies of MYC. The contrasting on the western blots makes it difficult to determine the degree of knockdown, but comparing the qPCRs in S3B and S6A, it appears that the knockdown is at least ~3X efficient in the TKO than in the parental cells. This affects interpretation of Figure 6, as less MYC knockdown in parental siMYC cells might naturally lead to less lost MYC-regulated genes. I understand that CRISPR knockout of MYC is likely impossible in these (and most) cells, and that siRNA is not easily tuned to a specific degree of knockdown, but the Authors should support their findings in Figure 6 with an additional method of MYC inhibition, such as expression of OmoMyc or use of a MYC-targeting pharmacologic agent such as MYCMI. A repeat of the RNA-seq experiment is not necessary; instead, repeating some of the individual genes in 6D would be sufficient.

* The mechanism of how TXNIP regulates MYC remains unclear. In the Discussion, the Authors state that, "Our preliminary experiments provisionally rule out a role for TXNIP in regulating global chromatin accessibility, the amount of Myc in the nucleus, the formation of Myc:Max heterodimers, or Myc's association with cofactors required for genome binding such as WDR5 (46); unpublished data)." Any of these results would be highly welcomed as negative results in the Supplemental Data section, particularly those connected to global chromatin accessibility. In particular, a change in chromatin accessibility upon TXNIP loss would be fully consistent with the idea of MYC as a global amplifier of transcription, while a lack of correlation between TXNIP loss and major changes in chromatin accessibility would be an interesting counterpoint to this model. If the Authors are not ready to show these data or other data mentioned in the Discussion in the paper, they should remove this sentence.

* I believe that Figure 6C is the most important panel in the entire paper, as it shows that the degree of expression regulation by MYC is not changed in the 231:TKO siMYC versus 231 parental siMYC despite the different numbers of MYC targets. However, this panel is somewhat difficult to read and interpret. In addition to 6C itself, it would be useful for the Authors to show these data another way, such as overlapping histograms showing degree of expression in the two models.

* For the same reason that it was important to include a proliferation assay for 231:TKO cells (Figure S1D), the Authors must include proliferation assays for other manipulations, such as the 231:TKO + siMYC cells and the parental 231 siMYC cells. This will help with evaluation of some of the RNA-sequencing results, particularly those related to cell cycle pathways, which may be regulated by MYC but also by proliferation state in general. A difference in growth between the 231:TKO + siMYC cells and the parental 231 siMYC might even suggest that the parental 231 siMYC are less able to tolerate MYC knockdown (see comment below about different degrees of siRNA efficiency). 

* There are several pieces of data in the paper that are lacking in an explanation of the Methods, these need to be included:

1) How TXNIP mRNA was used to rank the levels of all other genes in 1904 breast cancer lines, and to create a single pre-ranked list for GSEA analysis.

2) How GSEA results throughout the paper were identified for presentation. Were these based on rankings of NES strength, or hand-curated from a larger list of results? I am not opposed to the results being hand-curated since the Authors are focused on pathways related to MYC and metabolism, but this must be clearly stated.

Minor (non-essential) points:

* The Authors should discuss why MDA-MB-231 was chosen, particularly with regard to MYC status. Is MYC amplified in these cells? If not, is it highly expressed compared to other TBNC lines or other breast cancer lines? A supplemental graph using data from the Cancer Cell Line Encyclopedia / DepMap to show where MDA-MB-231 MYC expression falls in breast cancer cells lines would be useful. This information is important in evaluating the conclusions of the paper. Since not all cancers have elevated MYC, TXNIP deletion in cancers with closer to wild-type levels of MYC may perhaps not result in expanded MYC DNA binding, due to lack of excess MYC protein.

* In Figure 6, comparing 231:TKO siMYC and 231 parental siMYC, are there pathways that are enriched in 231:TKO siMYC but not in the parental? It might be useful to include a panel that outlines a few of these pathways, along with focusing on the change in enrichment in existing pathways in Figure 6E

* The western blot in Figure S1E should be rerun so that the parental and TKO are on the same blot, and can be uniformly contrasted.

Reviewer #3: The MYC protooncogene plays a important role in the regulation of metabolism including there regulation of glucose uptake to drive glucose dependent biosynthetic pathways. MYC regulates these processes through transcriptional regulation of genes including those that drive the expression of glucose transporters. Further, the authors previously have shown that MYC represses the expression of Thioredoxin Interacting Protein (TXNIP). This serves as a potent negative regulator of glucose uptake. They showed before that a Myc high /TXNIP low gene signature may be clinically significant because it appears to predict the overall clinical prognosis in Triple-Negative Breast Cancer (TNBC) but not in other subtypes of breast cancer. Now the authors have examined further the role of TXNIP function and how it may contribute to the aggressive behavior of TNBC. They generated TXNIP null MDA-MB-231 (231:TKO) cells. They found that TXNIP loss drives a transcriptional program that resembles those driven by Myc. They suggest that this happens because it increases global Myc genome occupancy. Thereby, they conclude that TXNIP loss allows Myc to invade the promoters and enhancers of target genes that are potentially relevant to cell transformation. They suggest that TXNIP is a broad repressor of Myc genomic binding. The increase in Myc genomic binding in the 231:TKO cells expands the Myc-dependent transcriptome. They argue that this leads to the expansion of Myc- dependent transcription following TXNIP loss occurs without an apparent increase in Myc's intrinsic capacity to activate transcription and without increasing Myc levels. Hence they suggest that TXNIP loss mimics Myc overexpression, connecting Myc genomic binding and transcriptional programs to the metabolic signals that control TXNIP expression.

Overall these results are potentially interesting but preliminary and provide only limited additional information compared to the authors previously published work. There is essentially data from only one cell line, in vitro, without a specific mechanism, or demonstration that the is a causal role in tumorigenesis or even in change in metabolism. There is only limited effort to validate that other potential mechanisms have been considered.

First, the experiments are essentially performed in one immortal human tumor derived cell line. Although the observations are compared to previously archived data, the main experiment is a single KO in a single cell line. This is a reasonable start, but to say that the results generally apply, or are specific to TNBC, is not justified based upon this data.

Second, the results are only performed using in vitro data. And, there is no experimentation to see if any of the effects on gene expression or effects of the KO have any effects on actual tumor biology, invasive behaviorr, dependence or changes in metabolism.

Some validation of observations using mouse models, xenografts, and/or other human-tumor derived cell lines should be included.

Third, the notion that MYC transcriptional effects are contextually dependent upon other gene products, metabolism, and/or transcriptional mechanisms has been well established by a multitude of investigators. The role of TXNIP and why this is novel beyond the authors prior work is not clear. Hence, the novelty and significance of the results needs to be clarified.

Fourth, the presumption that there are no effects on MYC expression are based upon limited data, such as a single Western blot that is not quantified. Seems very unlikely that there are no effects on MYC mRNA, protein expression stability. The constitutive KO is likely to have other effects on the cell line. 

Fifth, the mechanism proposed, that MYC promoter binding is globally enhanced as if MYC is overexpressed needs to be clarified. What is the mechanism? Do the authors think actually that a single gene KO always has the effect on MYC? Or only in this cell line? Or only in TNBC? is this a direct mechanism? How? How does this relate to the Levens and Young transcriptional enhancer model? Does this only explain "high" MYC then what is "high" versus physiologic? 

Overall, the authors make an interesting observation suggesting that KO of TXNIP influencing MYC transcription in one tumor cell line, but the results at this juncture are preliminary.

---

## [Editor Report · Decision Letter 2]

2 Feb 2023

Dear Dr Ayer,

Thank you very much for your patience while we considered your revised manuscript "TXNIP loss expands Myc-dependent transcriptional programs by increasing Myc genomic binding" for publication as a Research Article at PLOS Biology. This revised version of your manuscript has been evaluated by the PLOS Biology editors and by the Academic Editor.

Based on the reviews and our Academic Editor’s assessment of the revision, I am pleased to say that we are likely to accept this manuscript for publication, provided you satisfactorily address the following data and other policy-related requests that I provided below (A-G):

(A) We note that the new orthotopic xenograft data has been provided in the rebuttal but not in the manuscript. We ask that you please include this xenograft data in the paper and refer to the data in the manuscript text. 

(B) In the Methods section, please include the full name of the IACUC/ethics committee that reviewed and approved the animal care and use protocol/permit/project license. Please also include an approval number.

(C) You may be aware of the PLOS Data Policy, which requires that all data be made available without restriction: http://journals.plos.org/plosbiology/s/data-availability. For more information, please also see this editorial: http://dx.doi.org/10.1371/journal.pbio.1001797

-Supplementary files (e.g., excel). Please ensure that all data files are uploaded as 'Supporting Information' and are invariably referred to (in the manuscript, figure legends, and the Description field when uploading your files) using the following format verbatim: S1 Data, S2 Data, etc. Multiple panels of a single or even several figures can be included as multiple sheets in one excel file that is saved using exactly the following convention: S1_Data.xlsx (using an underscore).

-Deposition in a publicly available repository. Please also provide the accession code or a reviewer link so that we may view your data before publication.

Figure 1B-C, 2B, 2D-F, 3C-D, 4A, 4C, 4D, 4F-G, 5D, 6C-G, S1A-B, S1D-E, S2B-C, S2E-F, S4A, S4C-D, S5C-E, S6A, S6C-D, S7E, S8B-C

(D) Thank you for already depositing the RNA-seq, ChIP-seq and ATAC-seq data in the GEO. However, we note that this data is currently on hold and scheduled for release on July 17th 2023. We ask that you please ensure that this data is made publicly available before publication. 

(E) Please also ensure that each of the relevant figure legends in your manuscript include information on *WHERE THE UNDERLYING DATA CAN BE FOUND*, and ensure your supplemental data file/s has a legend.

(F) We require the original, uncropped and minimally adjusted images supporting all blot and gel results reported in the following figures:

Figure 1A, 2A, 4B, 4E, 6A, S1C, S2A, S2D, S10A-E

We will require these files before a manuscript can be accepted so please prepare and upload them now. Please carefully read our guidelines for how to prepare and upload this data: https://journals.plos.org/plosbiology/s/figures#loc-blot-and-gel-reporting-requirements

(G) Please ensure that your Data Statement in the submission system accurately describes where your data can be found and is in final format, as it will be published as written there.

We expect to receive your revised manuscript within two weeks. 

*Published Peer Review History*

*Press*

Kind regards,

Richard

Richard Hodge, PhD

Associate Editor, PLOS Biology

rhodge@plos.org

PLOS

---

## [Editor Report · Decision Letter 3]

13 Feb 2023

Dear Dr Ayer,

Thank you for the submission of your revised Research Article "TXNIP loss expands Myc-dependent transcriptional programs by increasing Myc genomic binding" for publication in PLOS Biology. On behalf of my colleagues and the Academic Editor, Connie Eaves, I am pleased to say that we can accept your manuscript for publication, provided you address any remaining formatting and reporting issues. These will be detailed in an email you should receive within 2-3 business days from our colleagues in the journal operations team; no action is required from you until then. Please note that we will not be able to formally accept your manuscript and schedule it for publication until you have completed any requested changes.

PRESS

Kind regards, 

Richard

Richard Hodge, PhD

Associate Editor, PLOS Biology

rhodge@plos.org

PLOS
